# Widespread correlation patterns of fMRI signal across visual cortex reflect eccentricity organization

Michael J Arcaro[1,2]*, Christopher J Honey[3], Ryan EB Mruczek[4], Sabine Kastner[1,2], Uri Hasson[1,2]

[1]Princeton Neuroscience Institute, Princeton University, Princeton, United States; [2]Department of Psychology, Princeton University, Princeton, United States; [3]Department of Psychology, University of Toronto, Toronto, Canada; [4]Department of Psychology, Worcester State University, Worcester, United States

**Abstract** The human visual system can be divided into over two-dozen distinct areas, each of which contains a topographic map of the visual field. A fundamental question in vision neuroscience is how the visual system integrates information from the environment across different areas. Using neuroimaging, we investigated the spatial pattern of correlated BOLD signal across eight visual areas on data collected during rest conditions and during naturalistic movie viewing. The correlation pattern between areas reflected the underlying receptive field organization with higher correlations between cortical sites containing overlapping representations of visual space. In addition, the correlation pattern reflected the underlying widespread eccentricity organization of visual cortex, in which the highest correlations were observed for cortical sites with iso-eccentricity representations including regions with non-overlapping representations of visual space. This eccentricity-based correlation pattern appears to be part of an intrinsic functional architecture that supports the integration of information across functionally specialized visual areas.

*For correspondence: marcaro@ princeton.edu

**Competing interests:** The authors declare that no competing interests exist.

## Introduction

Within the visual system, a prominent organizational principle is that of retinotopy: adjacent neurons along the cortical surface typically receive input from adjacent points on the surface of the retina. Each retinotopic map can be divided along two orthogonal axes: polar angle (i.e., angular distance) and eccentricity (i.e., radial distance). Early electrophysiological recordings in monkeys and cats identified several representations of the contralateral visual field in and around the calcarine sulcus (*Daniel and Whitteridge, 1961*; *Zeki, 1969*; *Allman and Kaas, 1971*; *Van Essen et al., 1984*). Through the use of functional magnetic resonance imaging (fMRI), it has now become evident that the human visual system contains over two-dozen visual maps (for review of original mapping studies, see *Silver and Kastner, 2009*; *Wandell and Winawer, 2011*; *Abdollahi et al., 2014*; *Wang et al., 2014*).

The topographic organization of individual areas is thought to provide an infrastructure for the integration of information across areas and along the visual hierarchy (*Kaas, 1997*; *Wandell et al., 2007*). Anatomical studies in primates have demonstrated that neurons with overlapping receptive fields (RFs) are interconnected (*Cragg, 1969*; *Zeki, 1969*; *Van Essen and Zeki, 1978*; *Maunsell and Van Essen, 1983*). Similarly, fMRI connectivity studies in humans have demonstrated topographically-local correlations between regions with overlapping visual field representations (*Heinzle et al., 2011*; *Haak et al., 2012*; *Butt et al., 2013*; *Donner et al., 2013*; *Gravel et al., 2014*; *Raemaekers et al., 2014*). In addition, widespread functional correlation patterns have been observed across visual cortex in macaques (*Leopold et al., 2003*; *Vincent et al., 2007*) and humans (*Nir et al., 2006*, *2008*; *Yeo et al., 2011*;

**eLife digest** Imagine you are looking out over a scenic landscape. The image you perceive is actually made up of many different visual components—for example color and movement—that are processed across many different areas in a region of the brain called the visual cortex. An important question for neuroscience is how the visual system combines information from so many different areas to create a coherent picture of the world around us.

Many areas of the visual cortex have their own map of what we see (the visual field). These maps allow the brain to maintain its representation of the visual field as the information passes from one processing area to the next. Areas that process corresponding parts of the visual field are physically interconnected, and tend to be active at the same time, which suggests that they are working together in some way. In addition, areas of the visual cortex that process different sections of the visual field can be activated at the same time, but it is not clear how this works.

Here, Arcaro et al. used a technique called functional magnetic resonance imaging (fMRI) to image the brains of people as they watched movies and while they rested. The images showed that seemingly unrelated areas of the visual cortex respond in similar ways if they are processing sections of the visual field that are the same distance from the center of the person's gaze. For example, if you look directly at the center of a computer screen parts of the brain that process the top of the screen are active at the same time as parts that process the bottom.

Arcaro et al.'s findings suggest that the brain uses the distance from the center of our gaze to bring together information from different areas of the visual cortex. This offers a new insight into how the brain assembles the many pieces of the visual jigsaw to make a complete picture. Future work will investigate how the architecture of the visual cortex is able to support this coupling of different areas, and how it might influence our perception of the visual world.

*Donner et al., 2013*). These patterns contain broad differences between foveal and peripheral cortex (*Raemaekers et al., 2014*), though may also be tied to the fine-scale organization of individual retinotopic maps.

In this study, we used fMRI to investigate the relationship between the spatial pattern of correlated BOLD signal and the retinotopic organization of eight visual areas (V1–hV4, VO1–2, V3A–B). Correlation analyses were performed on data collected during task-free conditions (eyes-closed and fixation), and during movie viewing. Correlation patterns were consistent across subjects and experiments. In addition to finding patterns that support well-established anatomical connectivity between areas with overlapping RFs, our analyses revealed a widespread correlation pattern based on eccentricity representation, in which the BOLD signal was correlated in areas with non-overlapping visual field representations, but with matching eccentricity representations. This eccentricity-based correlation pattern was observed between upper and lower visual field representations, within and across visual areas, and between hemispheres. Moreover, correlation patterns were similar in the presence and absence of bottom-up visual input. Finally, the measured correlation pattern could not be accounted for by overlapping RFs, inter-hemisphere homotopic connections, anatomical distance, eye movements, subject motion, or physiological noise. Our results demonstrate that functional coupling between visual areas reflect both local and widespread topographical patterns. We propose that this widespread pattern is part of an intrinsic functional architecture of the visual system that could reflect eccentricity-dependent processing.

## Results

Retinotopy and correlation patterns were characterized and compared within the visual systems of 14 participants. Retinotopic organization of the visual system was examined within the central 15° of visual space using a conventional travelling wave paradigm in which eccentricity and polar angle maps were collected to define visual areas V1, V2, V3, V3A–B, hV4, VO1–2 (see 'Materials and methods'). The organization of correlation patterns was probed in two resting conditions in which participants were instructed to either (1) keep eyes open and maintain fixation on a centrally presented dot or (2) keep eyes closed for the duration of the run. In addition, we assessed the correlation patterns in 11 of these participants during a naturalistic viewing condition in which participants were instructed to attend to a movie, but maintain fixation on a centrally presented dot.

## Exploratory seed analysis

To investigate the topography of functional correlations across visual cortex, Pearson correlation coefficients were computed between the timeseries of all surface data points (nodes) within the left and right visual cortices. For any individual node, correlation coefficients with all other nodes within visual cortex typically ranged between −0.10 and 0.75 in individual subjects (after cerebrospinal fluid and white matter signal regression). For illustration of the raw functional correlation results, we present fixation resting-state correlation maps for four example seed locations in subject S1 (*Figure 1A*), eyes shut resting-state correlation maps for four example seed locations in subject S2 (*Figure 1B*), and group average resting-state maps for four example seed locations (*Figure 2A*). In each of the four seed locations, the BOLD timeseries was sampled from a single node (black dot) within right dorsal V2, and the seed location was gradually shifted from foveal (<1.0°) to peripheral-most (~11.5°) representations as defined from a separate eccentricity localizer experiment (rightmost panel).

In all cases, we observed a combination of topographically local and widespread correlation patterns (with respect to visual field representations). Each of the four seed locations was strongly correlated (red / yellow) with adjacent cortex. In addition, the strongest correlations with each seed extended across visual cortex and spanned several visual areas, from V1 to V3A–B, dorsally, and to VO1–2 ventrally. Strong correlations with nodes adjacent to the seed and within ipsilateral dorsal cortex likely reflect well-established connectivity based on overlapping visual field representations (see *Heinzle et al., 2011*; *Haak et al., 2012*; *Butt et al., 2013*), but may also reflect the intrinsic spatial spread of the BOLD signal (*Engel et al., 1997*; *Parkes et al., 2005*). Interestingly, correlations were seen within *both* dorsal and ventral occipital cortex, comprised of lower and upper visual field representations respectively (*Figure 1*). Despite anatomical distance and representing a different part of visual space, the eccentricity representation (distance from fovea) of peak correlations within ventral occipital cortex corresponded to that of the seed location. These four seed locations also yielded comparable correlation patterns in the contralateral (left) hemisphere, comprised of right visual field representations. Local and widespread correlation patterns were observed in most individual subjects and the group average data for dorsal and ventral cortex seeds in V2 and V3, regardless of seeding near the horizontal or vertical meridians (see *Figure 1—figure supplements 1, 2* for additional individual subject data).

The BOLD signals in areas with eccentricity preferences similar to the seed were correlated in the presence and absence of visual input, even in cases where the spatial receptive fields (RFs) were non-overlapping (i.e., across lower and upper or right and left visual field representations). To summarize the group average V2-seeds correlation results, we projected the correlation maps in *Figure 2A* into visual field coordinates, and averaged across areas V1, V2, V3, V3A–B, hV4, and VO1–2 (*Figure 2B*). The correlation patterns highlight both visuotopically local and widespread correlation patterns (*Figure 2B*). Peak correlations (red) were evident in parts of the visual field around each seed location with strong correlations (red / yellow) also extending across the visual field along an eccentricity ring corresponding to that of the seed location. Similar local and widespread eccentricity-based correlation patterns were observed in data from the movie viewing experiment (*Figure 3*). Individual subject and group average correlation patterns were similar to previously reported group average correlation patterns (*Yeo et al., 2011*). Below, we formally tested the relation of eccentricity representations to the spatial pattern of correlated BOLD signal between individual brain areas and across tasks.

## Eccentricity binning

To characterize the widespread eccentricity-based correlation pattern that was observed in raw correlation maps, individual subject timeseries data were grouped by visual area and then partitioned into 12 bins between 0.50° and 12.50° of eccentricity. Data binning was used as a form of averaging to increase signal-to-noise within bins, while preventing the spread of signal between bins. Within-subject pairwise correlations were calculated between the mean timeseries of all bins for visual areas V1, V2, V3, hV4, V3A–B (combined), and VO1–2 (combined), each of which has sufficient surface area to allow for fine-scale binning of eccentricity data. Areas V1–V3 were separated into quadrants for most analyses. No additional extrastriate areas were included in these analyses.

Correlations between dorsal and ventral bins of V1, V2, and V3 were strongest for iso-eccentricity representations. These correlations were seen across the vertical and horizontal meridians at both foveal and peripheral-most bins. We first illustrate the binning analysis by examining the raw

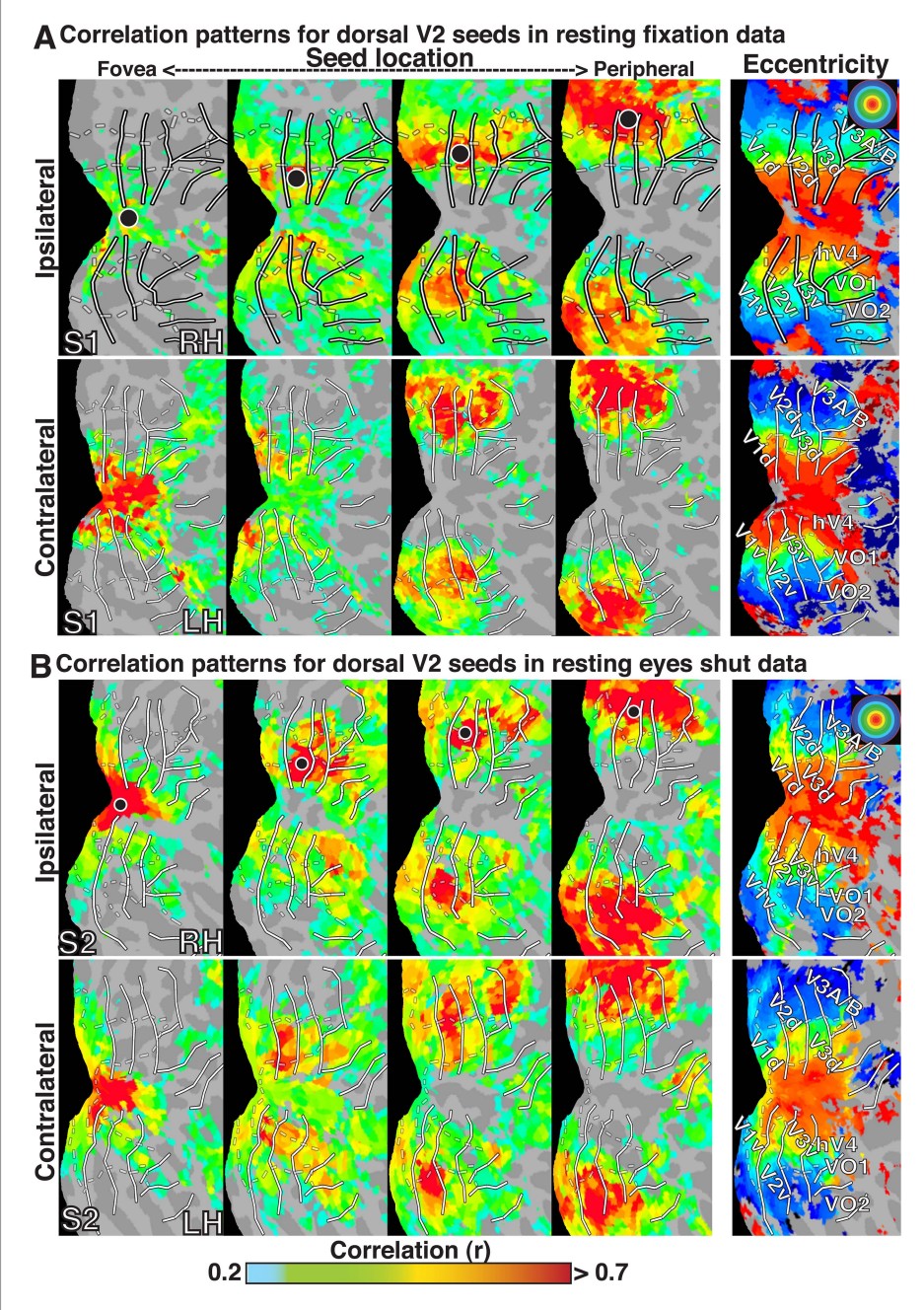

**Figure 1**. Seed-based correlations on resting state in individual subjects. Correlation maps in both hemispheres of (**A**) subject S1 for resting fixation and (**B**) subject S2 for resting eyes shut at four seed locations (<1.0°, ~2.5°, ~5.5°, ~11.5°; left to right) in dorsal V2 of the right hemisphere. For each seed, the strongest correlations (red / yellow) span several visuotopic areas within an eccentricity range roughly corresponding to that of the seed area (black dot) in both the ipsilateral and contralateral hemispheres. The correlations have a similar organization to eccentricity maps (far right). To facilitate visual comparisons between hemispheres, the left hemisphere images have been horizontally reflected. Solid white bars mark borders between visual field maps. White dashed bars outline three bands of iso-eccentricity.

The following figure supplements are available for figure 1:

**Figure supplement 1**. Seed-based correlations on resting state eyes shut in individual subjects.

**Figure supplement 2**. Seed-based correlations on resting state eyes shut in individual subjects.

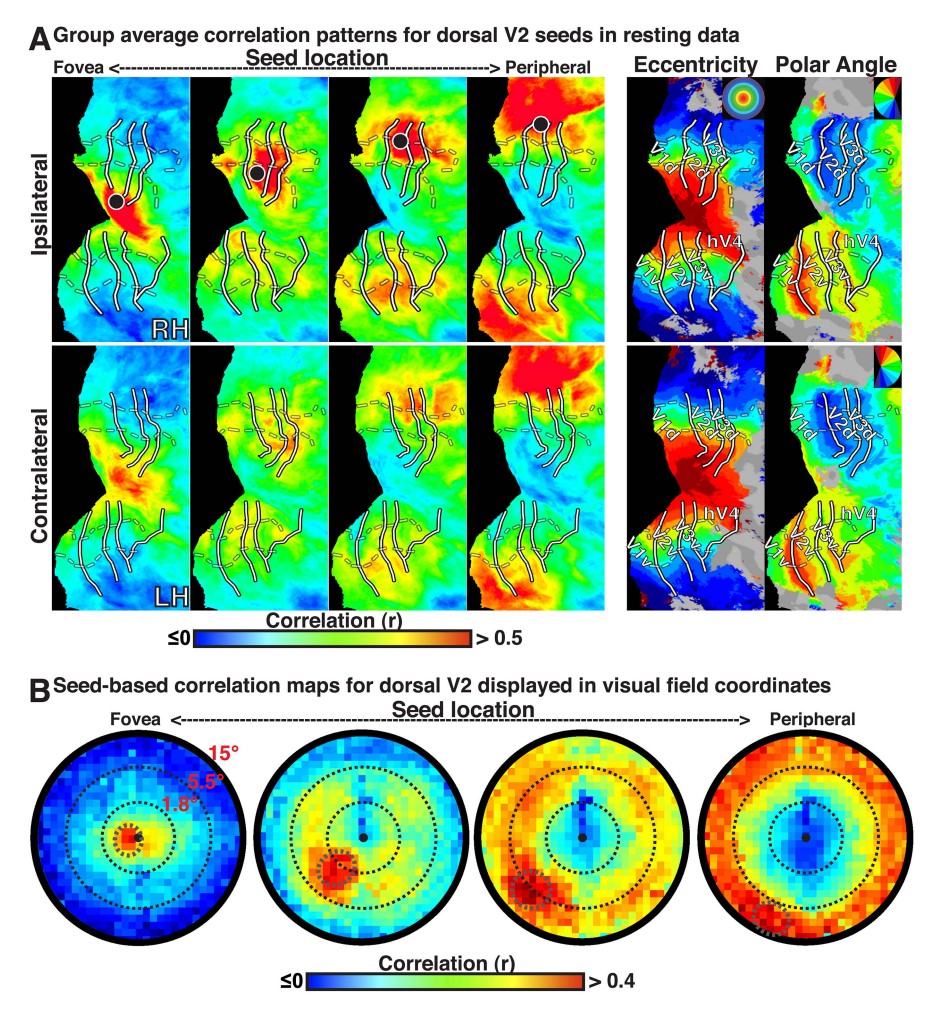

**Figure 2**. Group average seed-based correlations on resting state data. (**A**) Correlation maps in both hemispheres of group average data for resting fixation at four seed locations (<1.0°, ~2.5°, ~5.5°, ~11.5°; left to right) in dorsal V2. For each seed, the strongest correlations (red / yellow) span several visuotopic areas within an eccentricity range roughly corresponding to that of the seed area (black dot) in both the ipsilateral and contralateral hemispheres. The correlations have a similar organization to eccentricity maps (far right). To facilitate visual comparisons between hemispheres, the left hemisphere images have been horizontally reflected. Solid white bars mark borders between visual field maps. White dashed bars outline three bands of iso-eccentricity. (**B**) Seed-based correlations plotted as a function of visual field representation for four seed locations. Eccentricity values are derived from a log-scaled stimulus (see 'Materials and methods'). Black, dashed circles denote distance from fixation in visual degrees for each seed location.

correlations between the ventral and dorsal quadrants in V3 (*Figure 4*). This pair was chosen for illustration because correlations computed across quadrants are less susceptible to the influence of overlapping receptive fields (RFs) or cortical proximity. For regions that represent the same visual quadrant (e.g., the dorsal portions of visual areas V1, V2, and V3), it is difficult to dissociate effects due to shared eccentricity representations from overlapping receptive fields. Similarly, for adjacent dorsal regions, the shortest cortical distances are typically at corresponding eccentricity representations, making it difficult to dissociate eccentricity-related correlations from cortical (and volumetric) distance-based correlations. The dorsal and ventral portions of visual area V3 (as well as V2), however, only anatomically border each other at the fovea, represent different parts of the visual field (lower and upper, respectively), and thereby minimize both the overlapping receptive field and anatomical adjacency concerns. Thus, correlation analyses between these areas allowed us to test for widespread eccentricity-based correlation patterns in cases where effects of cortical

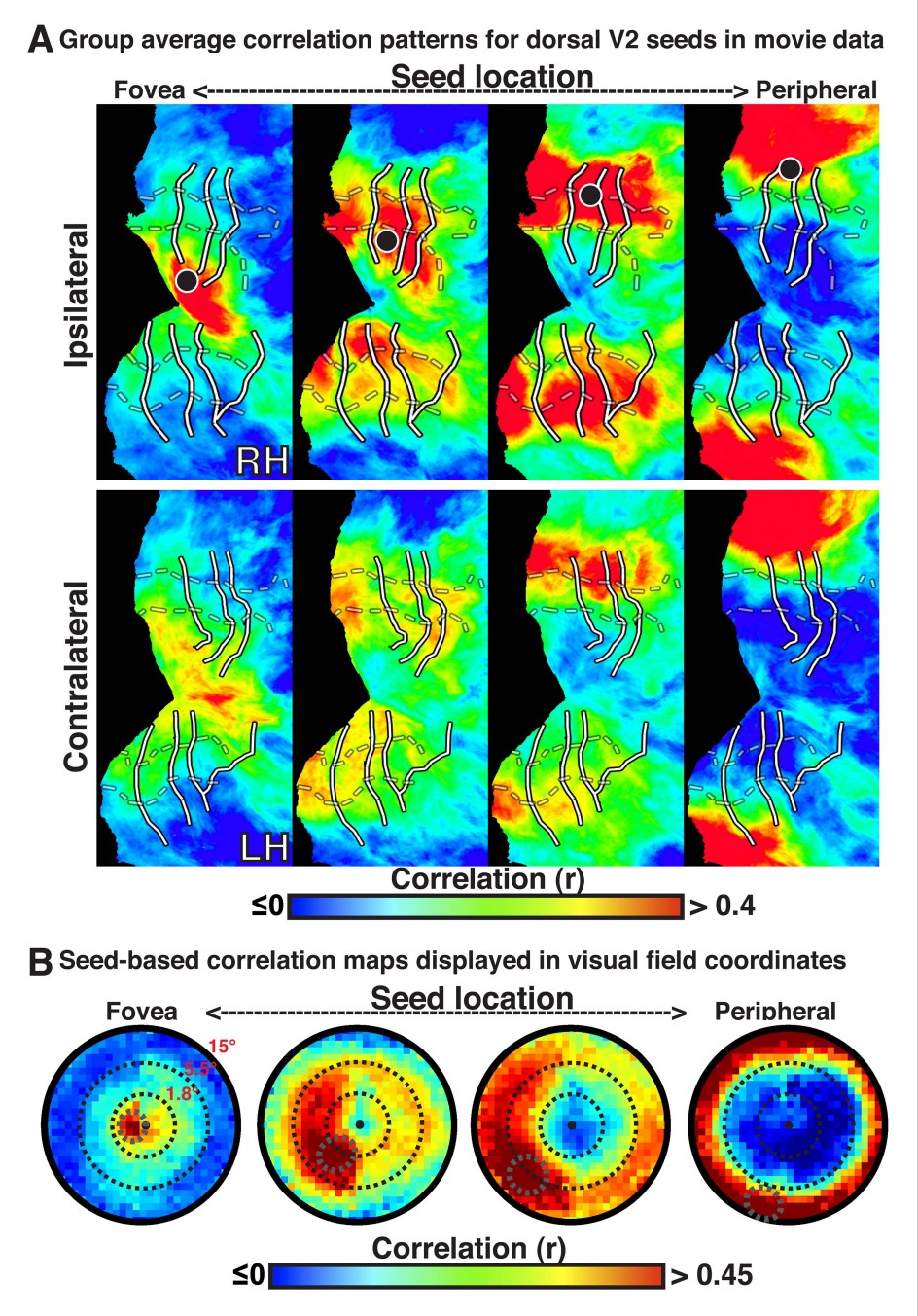

**Figure 3**. Group average seed-based correlations on movie viewing data. (**A**) Correlation maps in both hemispheres of group average data for movie viewing at four seed locations (<1.0°, ~2.5°, ~5.5°, ~11.5°; left to right) in dorsal V2. For each seed, the strongest correlations (red / yellow) span several visuotopic areas within an eccentricity range roughly corresponding to that of the seed areas (black dot). The correlations have a similar organization to the eccentricity maps (far right column of *Fig. 2*). Conventions the same as *Figure 2*. (**B**) Seed-based correlations plotted as a function of visual field representation for the four seed locations. Black, dashed circles denote the seed area. DOI: 10.7554/eLife.03952.007

distance and overlapping RFs are minimal. As seen for subject S4, the exploratory seed-based analyses showed strong correlations (red/yellow) between dorsal and ventral V3 at corresponding eccentricity representations (*Figure 4A*). Binned data showed the same correlation pattern. For each of the selected dorsal V3 bins, correlations with ventral V3 bins were strongest at

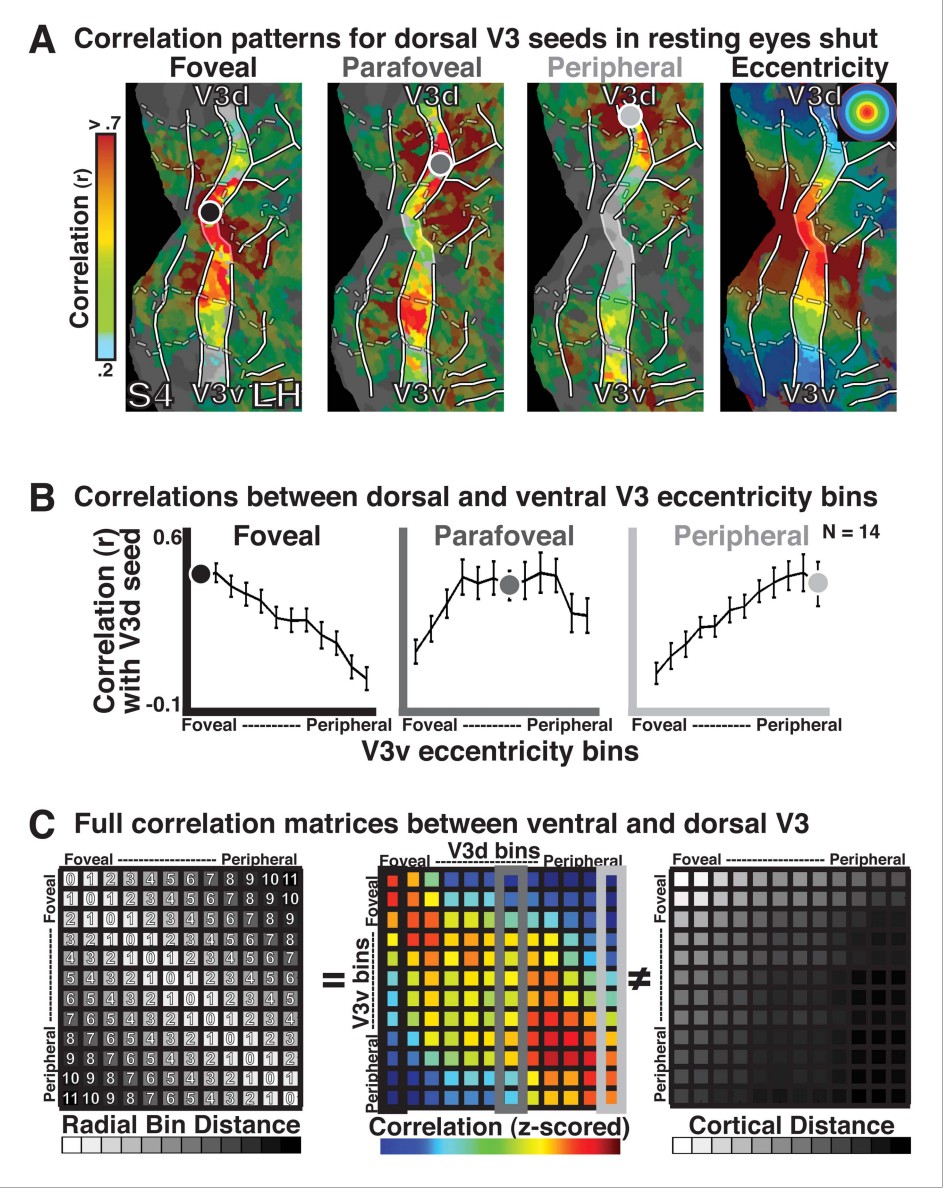

**Figure 4**. Illustration of eccentricity binning correlations on resting state. (**A**) Correlation maps in left hemisphere of subject S4 for 3 seed bin locations (<0.5–0.84°, 3.71–4.63°, 10.36–12.50°; left to right) in dorsal V3 and subject S4's eccentricity map (right). Grayscale dots mark approximate seed bin locations. (**B**) Correlations with all ventral V3 bins are plotted for the three dorsal seed locations. The strongest correlations between dorsal and ventral V3 were at, and around, iso-eccentricity representations. (**C**) The entire correlation matrix for all eccentricity locations between dorsal and ventral V3 (center) revealed a similar pattern where correlations were strongest at or near iso-eccentricity (i.e., the diagonal), and weaker for bins with large radial distances (e.g., foveal vs peripheral-most). Radial distance (left) was strongly correlated with the measured group connectivity (r = 0.84) and was uncorrelated to the cortical distance (right).

(and around) corresponding iso-eccentricities (*Figure 4B*). For example, peripheral-most V3d bins correlated most with peripheral-most V3v bins, and the correlations gradually decreased moving towards the foveal bin (*Figure 4B*, right). The entire correlation matrix for all eccentricity locations between dorsal and ventral V3 (*Figure 4C*, center panel), revealed a similar pattern, where correlations were strongest among ventral and dorsal bins with iso-eccentricity representations (i.e., the diagonal), and were weaker for bins with large radial distances (e.g., foveal vs peripheral-most).

The pattern of correlations reflects radial distance, not anatomical distance. We computed the (ranked) radial distance between eccentricity bins. A radial bin distance of zero corresponds to bins with the same (iso-) eccentricity representation, while a radial bin distance of 11 corresponds to bins furthest from each other on the eccentricity axis (i.e., foveal vs peripheral-most). *Figure 4C* provides the spatial pattern of correlations between ventral and dorsal quadrants in area V3 as well as the predicted patterns based on radial distance (left gray panel) and based on cortical distance (right gray panel). Note that both the radial bin distance and the cortical distance (as well as an overlapping RF model) predict strong correlations between foveal bins (*Figure 4C*). However, whereas cortical distance predicts that correlations will be weaker between cortically–distant iso-eccentricity bins (e.g., ventral and dorsal V3 peripheral-most), the radial bin distance predicts a strong correlation between iso-eccentricities (*Figure 4C*). Indeed, the group average bin data was strongly correlated with the predicted eccentricity pattern ($r = 0.84$) and was not positively correlated with cortical distance ($r = -0.04$) or volume distance ($r = -0.19$) (*Figure 4C*). Individual subject matrices were also correlated with radial distance (mean $r = 0.45$), and the Fisher-transformed correlations significantly differed from zero across subjects (one-sample t-test, $t(13) = 6.37$, $p < 0.0001$).

Correlations with radial bin distance were apparent within and across all visual areas tested, including areas with overlapping and non-overlapping visual field representations (both within and across hemispheres). We computed intra- and inter-hemisphere correlations as a function of eccentricity bin within and across visual areas V1, V2, V3, hV4, V3A–B (combined), and VO1–2 (combined) in each condition (resting eyes closed, resting eyes open, and movie fixation; *Figure 5*). In *Figure 5*, gray frames denote comparisons between regions with overlapping visual field representations while black frames denote comparisons between regions with minimal or no overlap in visual field representations (i.e., across ventral and dorsal visual fields and across hemispheres). For all comparisons in each condition the strongest correlations (red/yellow) were consistently between bins at comparable eccentricity locations (i.e., along the diagonal of each sub-matrix in *Figure 5*). This was the case even for correlations between dorsal and ventral visual portions of V1, V2, and V3, and between hemispheres, which represent mostly discrete parts of visual space. Across area pairs, the average individual subject correlation coefficient with radial distance ranged between 0.17 and 0.80 with a median of 0.39 and an interquartile range of 0.12. Fisher-transformed individual subject correlations significantly differed from zero for all pairs in each experiment (one-sample t-test, $t(13) > 2.31$, $ps < 0.05$, FDR-corrected) except for two pairs of inter-hemisphere correlations with hV4 in the movie viewing condition ($ps = 0.13$ and $0.07$). In all

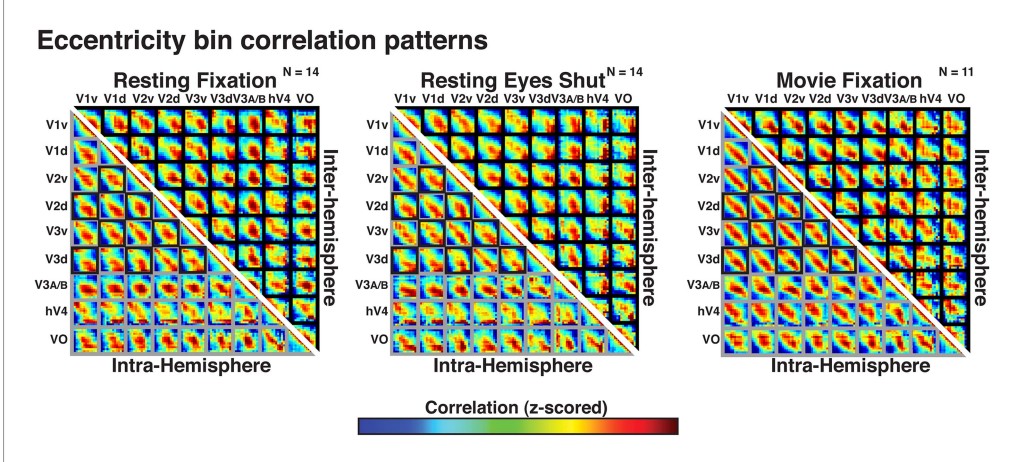

**Figure 5**. Correlation matrices for resting state and movie viewing conditions. Intra- (bottom-left matrix triangle) and inter-hemisphere (top-right matrix triangle) correlations are shown for all pairwise comparisons between visual areas V1, V2, V3, hV4, VO1-2, and V3A-B for resting fixation (left matrix), resting eyes shut (center matrix), and movie viewing (right matrix) experiments. For each pair of visual areas, the strongest correlations (red) are at corresponding eccentricity bins and its neighbors (diagonal in each sub-matrix). Grey and black boxes bound area pairs with overlapping and non-overlapping visual field representations, respectively.

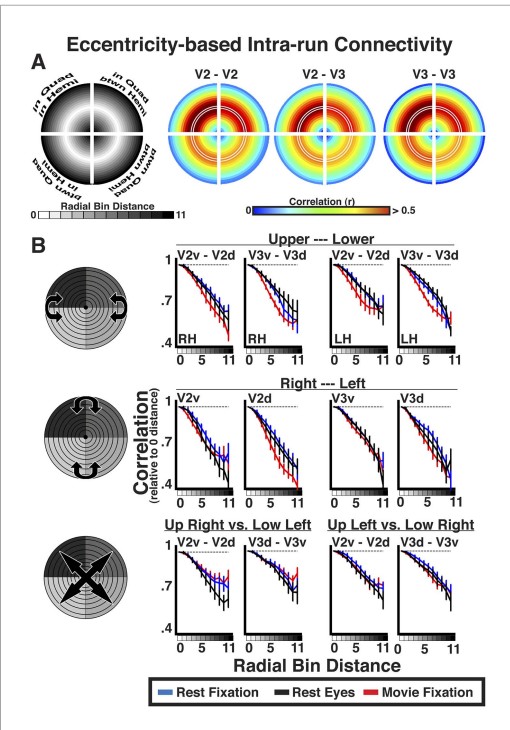

**Figure 6**. Intra-run eccentricity-based connectivity analyses for resting-state and movie viewing conditions. (**A**) Radial bin distance correlation plots between resting-state data bins for within quadrant/within hemisphere (upper left), within quadrant/between hemisphere (upper right), between quadrant/within hemisphere (lower left), and between quadrant/between hemisphere (lower right) comparisons. In each quadrant, the mid-radial arc (white outline) corresponds to the average correlation at iso-eccentricity with outer and inner arcs corresponding to average correlations at increasingly larger radial distances. (**B**) Average individual subject correlations are plotted for areas V2 and V3 as a function of radial distance between the upper and lower visual fields (top), right and left visual fields (middle), as well as across both left and right and upper and lower visual fields (bottom) for all conditions. Correlations are normalized to the average correlations at iso-eccentricity (0-radial distance). All correlations were strongest at the 0-radial distance and steadily decreased at larger distances for all conditions.

The following figure supplements are available for figure 6:

**Figure supplement 1**. Intra-run eccentricity-based connectivity analyses for resting-state and movie viewing conditions without meridian data.

**Figure supplement 2**. Intra-run angular-based connectivity analyses for resting-state and movie viewing conditions.

**Figure supplement 3**. Real data vs artificial data.

experiments, individual subject correlations with radial distance were generally strongest for early visual areas (V1, V2, and V3). This could reflect the relatively smaller, more spatially focal receptive fields in early visual areas, but also the larger surface area (i.e., more data points).

## Effect of radial distance

For each area pair, correlation coefficients were positive at 0–radial distance, and decreased at larger radial distances across the visual field. To assess the general effect of radial distance for each area pair, eccentricity bin correlations were grouped as a function of radial bin distance and averaged. Averaged correlations were plotted relative to iso-eccentricity (*Figure 6A*, left panel). In each quadrant, the radial mid-arc (*Figure 6A*, white outline) corresponds to the average correlation of the 0-distance bin (iso-eccentricity). Radial arcs further away from the mid-arc correspond to larger radial distances. For each radial plot, for example, V2–V3, radial arcs inside of the mid-arc illustrate correlations between relatively more foveal representations of V2 and relatively more peripheral representations of V3. Radial arcs outside of the mid-arc illustrate correlations between relatively more peripheral representations of V2 and relatively more foveal representations of V3. Inner and outer radial points will be identical for within area comparisons (e.g., V2–V2), but could differ for between area comparisons (e.g., V2–V3). As illustrated by resting state correlations within and between areas V2 and V3 (*Figure 6A*), iso-eccentricity bins were positively correlated, regardless of visual field quadrant. This was observed in all experiment conditions and area comparisons, and individual subject Fisher-transformed correlations significantly differed from zero ($ps < 0.05$, FDR corrected). There were clear differences in overall correlation magnitude between quadrants. As expected, correlation coefficients decreased at larger radial distances for foveal and peripheral-most bins in each quadrant, regardless of this magnitude difference (*Figure 6A*).

The relative decrease in correlation strength as a function of radial distance was similar across the visual field and across experiments (*Figure 6B*). We plotted the average correlation as a function of radial bin distance (relative to iso-eccentricity; see 'Materials and methods') between the upper and lower visual fields, right and left visual fields, as well as across both left and right and upper and lower visual fields separately (see graphic illustrations in *Figure 6B* left panels). Data are presented for comparisons between quadrants within and between areas V2

and V3. As discussed in the eccentricity binning section, these comparisons were chosen for illustration because overlap in visual field representation is minimal. Consistent with the radial arc plots (*Figure 6A*), correlations between dorsal and ventral portions of V2 and V3 were strongest at 0–radial distance, and steadily decreased at larger radial distances for all conditions (*Figure 6*; top row). We observed similar correlation patterns as a function of radial bin distance across hemispheres, both along the horizontal plane (e.g., RH V2v and LH V2v; middle row) and diagonally across the upper and lower visual fields (e.g., RH V2v and LH V2d; bottom row). The patterns were observed even after removal of horizontal and vertical meridian data, further demonstrating that these effects are unlikely to be driven by local, overlapping visual field representations (*Box 1*; *Figure 6—figure supplement 1*). The decrease in correlation coefficients at larger radial bin distances yielded negative slopes from linear fits in each subject for all V2 and V3 pairs. For any given pair, negative slopes were similar across condition. Across V2 and V3 pairs, the average individual subject slope of the linear fit ranged between −0.22 and −0.49. Further, the average individual subject slope for each area pair was greater than 97.5% of a permuted distribution (mean 97.5% across areas, conditions = −0.06) where the labels of radial distances were scrambled before deriving individual subject radial distance correlations and slopes. As seen in the radial plots (*Figure 6B*), the slopes were similar across regional comparisons, though were slightly shallower for comparisons that spanned both quadrants and hemispheres (i.e., diagonal; bottom row). Finally, we observed these effects during resting fixation and eyes closed conditions, as well as during the processing of the movie, attesting to the robustness of the effect, and excluding many potential confounds (see 'Control analyses'). Similar results were found in all other comparisons between visual areas. The magnitude of correlation coefficients linearly decreased as a function of radial bin distance for all area comparisons (V1, V2, V3, hV4, VO1–2, V3A–B), though slopes were generally steeper for comparisons between V1, V2, and V3. Across all areas, average individual subject slopes ranged between −0.14 and −0.80, and each was greater than 97.5% of the permuted distribution (mean 97.5% across areas, conditions = −0.09).

## Box 1. Widespread connectivity is not dependent on meridian representations.

Though we divided data into quadrants that represent distinct parts of visual space, monosynaptic connections have been observed between neurons in dorsal and ventral visual cortex with receptive fields overlapping at the horizontal meridian (*Jeffs et al., 2009*; see also; *Zeki, 1971*; *Stepniewska and Kaas, 1996*; *Felleman et al., 1997*; *Gattass et al., 1997*) and between hemispheres at the vertical meridian (*Essen and Zeki, 1978*; *Newsome and Allman, 1980*; *Cusick et al., 1984*; *Kennedy et al., 1986*; *Abel et al., 2000*). Despite constituting (at most) a minimal portion of the signal in our bin data, we tested whether the observed eccentricity-based correlations between dorsal and ventral cortex and between the hemispheres were driven by the inclusion of these overlapping representations at the horizontal and vertical meridians, respectively, by removing data from the meridians and re-running the binning analyses. For correlations between upper and lower visual fields (dorsal and ventral cortex), we cut out 60° of polar angle centered on the horizontal meridian, which spared 60° of polar angle in the upper and lower visual fields near the vertical meridians. For correlations between the right and left visual fields (right and left hemispheres), we cut out 90° of polar angle centered on each vertical meridian, which sparred 90° of polar angle centered on the horizontal meridian in each hemisphere. Consistent with the original analyses, average individual subject correlations between dorsal and ventral portions of V2 and V3 as well as between right and left hemispheres were strongest at the 0–radial distances (iso-eccentricities), and linearly decreased at larger distances for all conditions (*Figure 6—figure supplement 1*). In general, the slopes of correlations were marginally shallower, but did not significantly differ from the full data reported in *Figure 5* (*p*s > 0.05, FDR corrected). If overlapping meridian representations facilitated the correlation patterns, removal of this data should have eliminated the correlation effects. These data further support the interpretation that the observed correlations with radial distance are not due to overlapping representations of visual space.

## Effect of angular distance

For comparisons between dorsal areas, between ventral areas, and between mirror-symmetric (across vertical meridian) areas, correlations were positive at 0–angular distance, and decreased at larger angular distances (*Figure 6—figure supplement 2*). To assess the general effect of angular distance for each area pair, data were grouped into 12 equally spaced polar angle bins. Correlations between bins were computed and grouped as a function of angular bin distance. For within-quadrant, within-hemisphere and mirror symmetric (across vertical meridian) within-quadrant, between-hemisphere comparisons, correlations steadily decreased at larger angular distances. Taken together, these results suggest that angular connectivity reflects overlapping RF between areas and mirror-symmetric connections between hemispheres. No effect of angular distance was observed between dorsal and ventral comparisons based on actual angular distance or when reflecting across the horizontal meridian (i.e., mirror symmetry). As with any negative finding, we cannot conclude with certainty that such angular-based connectivity does not exist, though, to our knowledge, no study has shown angular connectivity between regions with non-overlapping RFs.

## Stimulus-dependent and independent correlations

For each area pair, the spatial pattern of correlations (across eccentricity bins) was similar between experimental conditions. Across all areas, the average individual subject correlation of these spatial patterns between experimental conditions was 0.86 for intra-hemisphere comparisons and 0.75 for inter-hemisphere comparisons. The average individual subject correlation between resting-state and movie viewing conditions was 0.84 for intra-hemisphere comparisons and 0.71 for inter-hemisphere comparisons. Overall, these data are consistent with the exploratory seed based analyses, and suggest that the spatial pattern of correlations between visual areas tested were similar in the presence and absence of a strong bottom-up input.

Given the similarity of correlation matrices across all three conditions, this widespread eccentricity-based correlation pattern appears to reflect an eccentricity bias that is inherent to the organization of the visual system (i.e., stable during rest) and may support processing during active perception of the visual environment (e.g., during movie viewing). We conducted correlations between individual runs to directly test whether the patterns observed during the movie viewing condition reflected processing of the stimulus input. Intrinsic neural dynamics during the resting and movie conditions that are not related to the processing of visual stimuli, as well as non-neuronal artifacts (e.g., respiratory rate, motion), can only influence the pattern of correlations within each run, but should not induce correlations between runs. Indeed, inter-run Fisher-transformed correlations on resting state data showed no effect of radial distance (all $ps > 0.05$), validating the assumption that noise correlations should not be reliable across runs (*Figure 7*, blue and black lines). In contrast, inter-run correlations during movie watching showed radial distance effects similar to intra-run correlations (*Figure 7*, red lines). Inter-run correlations were statistically significant for 96% of all comparisons ($ps < 0.05$, FDR corrected). The slopes of coefficients as a function of radial distance for movie data were, generally, weaker

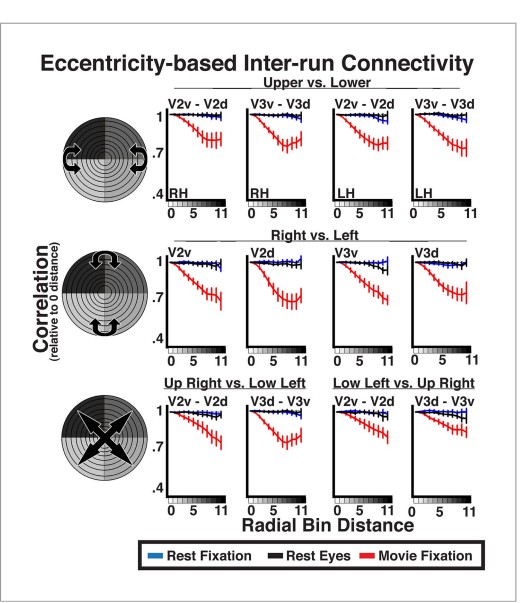

**Figure 7**. Inter-run eccentricity-based connectivity analyses for resting-state and movie viewing conditions. For the movie viewing condition, average individual subject inter-run correlations between hemispheres as well as between dorsal and ventral portions of V2 and V3 were strongest at the 0-radial distance (iso-eccentricity) and steadily decreased at larger distances. For resting state data, correlations did not vary as a function of radial distance. See *Figure 6* for conventions.

for the inter-run relative to intra-run analyses (compare *Figures 6B, 7*). However, the inter-run analyses inherently had less data and spatial attention was not controlled, making the differences between intra- and inter-run analyses difficult to interpret. The inter-run results indicate that the observed widespread eccentricity-based correlations can reflect the processing of the incoming information during viewing of real life stimuli, and further suggest that these correlations patterns are unlikely to be driven by non-neuronal artifacts.

## Control analyses

We performed a variety of control analyses to rule out possible confounds. The observed widespread eccentricity-based correlation pattern could not be attributed to non-neuronal artifacts (e.g., physiological noise, motion, BOLD spatial autocorrelation, eye movements). Global artifacts did not drive our results as correlation patterns were strengthened by the removal of non-neuronal signals (i.e., motion and white matter). The eccentricity-based correlation pattern was not due to inherent spread of the BOLD signal, anatomical distance, or any biases in our analyses, as artificial data that preserved such biases were not correlated with radial distance (see *Box 2*; *Figure 6—figure supplement 3*). Subject motion also could not explain the observed correlation patterns. We minimized the overall influence of movement in our data by using highly trained MRI subjects and by removing signal correlated with subject motion via regression. Further, the strength of eccentricity correlation did not significantly co-vary with degree of subject movement. We performed Spearman rank correlations between each subject's total amount of movement and the slope of his or her radial bin distance coefficients (i.e., effect of eccentricity on correlation strength). There was no significant correlation across subjects between the mean slope (averaged across all area pairs) and any of the six motion parameters (all rs < 0.24, $ps > 0.05$). In addition, there were no significant correlations across subjects between the slope and total amount of movement for any individual area pair ($ps > 0.05$, FDR-corrected). These analyses strongly suggest that the observed correlation patterns were not driven by subject motion. It is unlikely that eye movements drove the eccentricity-based correlation pattern as similar results were observed at rest during both fixation and eyes closed conditions. Finally, significant inter-run eccentricity-based correlation patterns were observed during movie viewing, but not during rest, further ruling out the possible influence of non-neuronal intrinsic artifacts (e.g., respiration rate, cardiac rate, motion), which should not be correlated across runs.

## Box 2. Widespread connectivity is not dependent on instrumentation and analyses.

The eccentricity-based correlation could not be accounted for by overlapping receptive fields or anatomical distance. There is an intrinsic spatio-temporal point-spread function in BOLD imaging (*Engel et al., 1997*; *Parkes et al., 2005*; *Shmuel et al., 2007*). The point-spread function of BOLD imaging at 3T had been estimated to be about 3.5 mm (*Engel et al., 1997*). We tested whether the observed correlation patterns were due to inherent spatial autocorrelation of the BOLD signal or pre-processing steps in our analysis pipeline by generating artificial data that maintained these non-neuronal spatial correlation patterns. We replaced the timeseries in each voxel with a randomly generated timeseries to preserve the anatomical position of each voxel in each subject and then applied a Gaussian spatial filter with a FWHM of 3.5 mm. We passed these data through the same processing as the real data (using motion correction parameters from each subject's real data) to derive estimated 'instrumental' correlations for each subject. These data capture the inherent spatial blur of the BOLD signal and preserve (any) volumetric and surface-based spatial correlations that were inherent in the data or introduced via the preprocessing and binning analyses. We found significant effects of radial distance for intra-areal correlations (e.g., V2d RH with itself; $ps < 0.05$) and weaker, though non-significant, effects for adjacent areas (e.g., V1d RH with V2d RH) (*Figure 6—figure supplement 3*). As discussed in the exploratory seed analysis section above, this was to be expected since the organization of eccentricity representations in these areas is correlated with (anatomical) spatial distance, and the spatial blurring in this analysis will introduce an anatomically local correlation structure to the data. Interestingly, the slopes of coefficients

across eccentricity disparities did not reflect the slopes observed in our real data, suggesting that this cannot account for the entire signal measured between these areas in our real data (*Figure 6—figure supplement 3*). For intra-area correlations, coefficient values sharply decreased as a function of radial distance, and resembled the exponential decay reported by Butt and colleagues in their simulations of instrumental correlations for visual area V1 (*Butt et al., 2013*). For correlations between adjacent areas, coefficient values decreased gradually for the three smallest distance values then leveled off at larger disparities. Again, this small decrease likely reflects anatomical distance, and the profile of distance correlations differed from the correlation pattern measured in our real data. Critically, there was no significant difference in correlation strength across distance values between dorsal and ventral regions (e.g., V2d RH to V3v RH; $ps > 0.05$) or across hemispheres (e.g., V2d RH to V2d LH; $ps > 0.05$) (*Figure 6—figure supplement 3*). We also scanned a phantom (using the same acquisition parameters as the resting data) to generate artificial data with the BOLD spatial autocorrelation, and found the same effects as with the randomly generated timeseries data. This control analysis demonstrates that the inherent (spatial) spread of BOLD imaging and anatomical distance cannot account for the observed correlations between areas with non-overlapping receptive fields (e.g., V2d RH to V2v RH or V2d RH to V2d LH), which we have focused on thus far (*Figures 4, 6, 7*). If anything, this control analysis suggests that it is important to account for BOLD spatial autocorrelation when evaluating correlation patterns within individual areas (e.g., V1d RH to V1d RH) and between (anatomically) adjacent areas (e.g., V1d RH to V2d RH).

## Topographic model regression analyses

Both local and widespread connectivity patterns contributed to the spatial pattern of observed correlations. Each visual hemifield map was separated into 36 bins: six divisions of eccentricity (E1–E6) that each contained six divisions of polar angle (A1–A6) such that each bin represented a unique part of visual space (*Figure 8—figure supplement 1*). To visualize the pattern of correlations across all bins with respect to local (overlapping RF) and widespread (eccentricity and polar angle) connectivity patterns, data were plotted as a function of radial and angular distance (*Figure 8*). Correlations were qualitatively strongest at the intersection of iso-polar angle and iso-eccentricity, and were generally elevated along iso-eccentricity representations, irrespective of polar angle distance (*Figure 8*). To quantify this, we assessed the relationship between the measured correlations across visual areas V1, V2, V3, and hV4 and several possible connectivity patterns using linear, least squares regression (see 'Materials and methods'). For any pair of visual areas, we modeled correlations between all bin pairs (36 × 36) as the linear weighted sum of four sources of connectivity. Two sources reflect topographically local connectivity between regions that are in close anatomical proximity or contain overlapping receptive fields: (1) instrumental 'noise' connectivity: correlation pattern based on an assessment of the spatial auto-correlation and preprocessing in our data; (2) overlapping RF connectivity: correlation pattern based on the overlap of estimated population receptive fields (pRF; *Dumoulin and Wandell, 2008*; *Amano et al., 2009*; *Harvey and Dumoulin, 2011*). The two other sources reflect topographically widespread connectivity that span much of the visual field and are not specifically tied to receptive fields or anatomical proximity: (1) eccentricity connectivity: correlation pattern based on radial distance with correlations strongest at iso-eccentricity representations; (2) polar angle connectivity: correlation pattern based on angular distance with correlations strongest at iso-polar angle representations.

In general, the strongest overlapping RF effects were observed for intra-areal comparisons whereas eccentricity effects were consistently observed across all comparisons and were much larger than overlapping RF and polar angle effects for inter-hemisphere comparisons. To quantify the apparent connectivity patterns, we conducted an initial regression analysis using a model that included all four sources of connectivity as predictors. For individual subjects, the combination of local and widespread predictors well fit intra-area correlation patterns, and moderately fit inter-area and inter-hemisphere patterns. The average individual subject variance explained by the full model ranged between 40% and 68% for intra-areal comparisons and between 11% and 28% for inter-areal comparisons. Across area pairs, there were consistent effects of both local and widespread predictors (*Figure 9—figure supplement 1*). Across subjects, the coefficients for the overlapping RF predictor

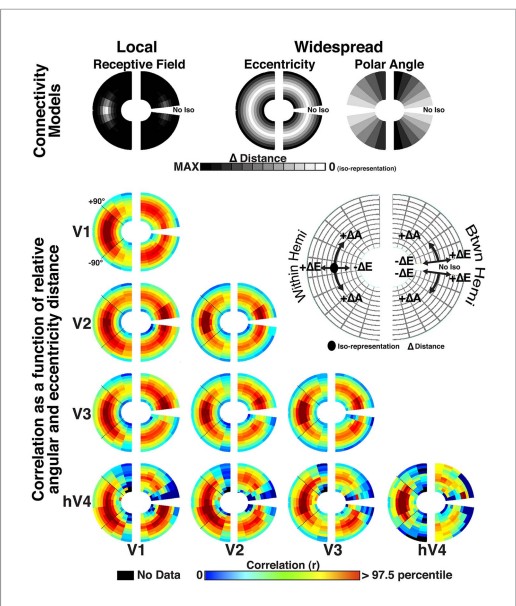

**Figure 8**. Radial and angular distance plots. Resting state correlation bins plotted as a function of average radial and angular distance (ΔE and ΔA, respectively) for areas V1, V2, V3, and hV4. Intra- and inter- hemisphere correlations are plotted in the left and right hemifields, respectively. For intra-hemisphere correlations (left hemifield), the mid arc (oval; see legend) corresponds to the average correlation at the intersection of iso-eccentricity and iso-polar angle. There is no iso-polar angle for inter-hemisphere comparisons (right hemifield), so plotted data begin off the horizontal meridian at an iso-polar distance of one, and correspond to the correlations between bins adjacent to the vertical meridian. Outer and inner arcs correspond to larger radial distance correlations (same conventions as symmetry plots in **Figure 6A**). Arcs closer to the vertical meridian correspond to larger angular distances (see legend). For each plot, for example, V2–V3, arcs above the mid arc (+ΔA) illustrate correlations between relatively more upper visual field representations of V2 and relatively lower visual field representations of V3 (reverse for below mid arc, −ΔA). For comparisons with area hV4, there was no data for some radial and angular distances (black arcs).

The following figure supplements are available for figure 8:

**Figure supplement 1**. Connectivity patterns for model fitting.

**Figure supplement 2**. Homotopic radial and angular distance plots.

were significantly positive for all intra-hemisphere, intra-areal comparisons as well as for most other comparisons ($ps < 0.05$, FDR-corrected). Coefficients for the eccentricity predictor were significantly positive for all but one comparison (V1–hV4 inter-hemisphere) ($ps < 0.05$, FDR corrected). Coefficients for the polar angle predictor were significantly positive for most intra-hemisphere comparisons, but only for a few inter-hemisphere comparisons ($ps < 0.05$, FDR corrected). Though these results suggest some effect of both local and widespread connectivity, we found that both eccentricity and polar angle predictors were strongly correlated with the overlapping RF predictor for most intra-hemisphere area pairs (mean $r = 0.54$; STD = 0.08). Since the local and widespread predictors have a non-trivial degree of collinearity, the coefficient estimates for individual predictors from the full model may not be accurate for intra-hemisphere comparisons, which had the strongest correlations between local and widespread predictors. From these results, it is possible that some of the observed eccentricity and polar angle effects are due to shared variance with the local, overlapping RF model. However, it is very unlikely that all of the widespread effects can generically be explained by the overlapping receptive field model since significant correlations were observed in the eccentricity bin analyses between iso-eccentricity representations in distinct parts of visual space (**Figure 6**; e.g., between V2d RH and V2v LH).

When removing the shared variance between local and widespread connectivity predictors, significant effects of eccentricity connectivity were still observed. To assess the contribution of widespread connectivity on the measured correlation patterns while controlling for the shared variance with the local connectivity model, we first removed the variance explained by local connectivity (i.e., overlapping RF and instrumental predictors), then calculated correlations between the residuals and the widespread connectivity models in each subject. The average correlation coefficient between the residual pattern and the eccentricity predictor was similar across most areas for both within and across hemispheres (**Figure 9**, medium gray bars). The Fisher-transformed individual subject eccentricity residual correlations were reliably different from zero for 56/60 visual area pairs across rest and movie viewing experiments ($ps < 0.05$, FDR corrected). Consistent with the binning analyses, residual correlations that included hV4 were weaker than residual correlations between areas V1, V2, and V3. In contrast, there was no clear, consistent effect of the polar angle predictor across areas (**Figure 9**, lightest gray bars). The polar angle residual correlations were

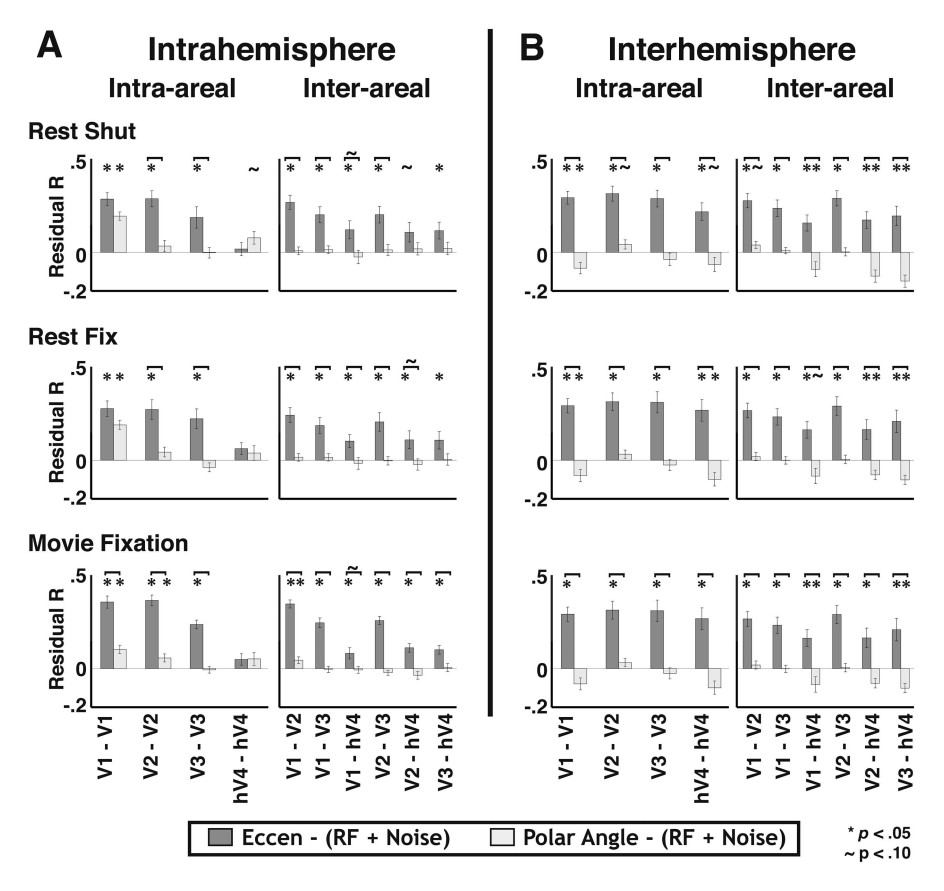

**Figure 9**. Residual correlations with eccentricity and polar angle predictors. (**A**) Intra- and (**B**) inter- hemisphere average individual subject correlations between the unexplained variance from an overlapping RF model fit and eccentricity (medium gray bars) and polar angle (light gray bars) predictors are plotted for all pairs of visual areas V1, V2, V3, and hV4. Residual correlations were significantly above 0 for 56/60 eccentricity comparisons (ps < 0.05; FDR corrected; one-sample t-test) and were significantly greater than polar angle correlations for 49/60 comparisons (ps < 0.05, FDR corrected; paired t-test). Correlations between the residuals and the eccentricity predictor were comparable within and across areas, as well as within and across hemispheres, though were generally weaker for comparisons with hV4. Notations above each bar denote significance relative to the null hypothesis (one-sample t-test) and brackets denote significant differences between conditions (two-sample t-test). * ps < 0.05; ~ps < 0.10 (FDR corrected).

The following figure supplements are available for figure 9:

**Figure supplement 1**. Intra- and inter- hemisphere effects of local and widespread connectivity.

**Figure supplement 2**. Residual correlations with eccentricity and polar angle predictors after accounting for potential effects of BOLD signal spread.

**Figure supplement 3**. Effect of homotopic connectivity.

only significantly positive for 4/60 pairs (ps < 0.05, FDR corrected). For all other comparisons, residual polar angle correlations tended to be negative. The residual pattern was more correlated with the eccentricity predictor than with the polar angle predictor for 49/60 pairs (ps < 0.05, FDR corrected) (*Figure 9*).

These residual widespread effects were robust to scaling of pRF sizes in the local connectivity model. We used the average pRF values reported across several studies to derive our model of overlapping RF connectivity, however the size of pRFs for some areas varied as much as 2× across these studies. To make sure that the variance in pRF estimates across studies didn't significantly affect

the results, we re-ran the model analyses using pRF estimates from individual studies to generate the model of overlapping RFs (see 'Materials and methods'). Using these parameters to derive the overlapping RF model had no appreciable effect on the magnitude or significance of residual correlations. Further, the magnitude and significance of the eccentricity-based effects were also largely unchanged when we scaled pRF estimates to account for any additional effects of the point spread function in the BOLD signal not captured in the original pRF estimate (see 'Materials and methods'; *Figure 9—figure supplement 2*). Overall, these control analyses provide additional evidence for a widespread eccentricity-based connectivity that cannot be accounted for by local connectivity patterns.

## Eccentricity effect does not reflect asymmetry in local connectivity patterns

Widespread eccentricity-based connectivity effects were consistently observed before and after removal of local connectivity variance. Even when attributing all shared variance between local and widespread predictors to the local predictor, eccentricity-based effects were still observed. In contrast, consistent polar angle effects were generally only observed prior to removal of local connectivity variance. Eccentricity effects were generally stronger than polar angle both before and after removal of local connectivity variance, suggesting that this difference is driven by asymmetries in widespread, not local, connectivity patterns. In further support of this, the variance explained by the local connectivity model was strongly correlated with both eccentricity (mean r across areas = 0.50; STD = 0.08) and polar angle (mean r across areas = 0.49; STD = 0.07) predictors, and there was little difference between them (mean *r* difference across areas = 0.01; STD = 0.06). Taken together, these data suggest that connectivity effects based on overlapping RFs are comparable along eccentricity and polar angle axes (i.e., isotropic with respect to visual field), which is consistent with anatomical studies in macaques that suggest visual field coverage of local, lateral connections is generally isotropic (*Angelucci et al., 2002*). Importantly, the pattern of widespread correlated BOLD signal beyond overlapping receptive fields reflects eccentricity organization.

Our data suggests little effect of polar angle connectivity beyond overlapping RFs. As with any negative finding, we can only report that we failed to see an effect of angular connectivity between regions with non-overlapping RFs, but do not conclude with certainty that it does not exist. It is important to note that significant polar angle correlations were observed in the full model analysis for several area pairs. However, these correlations were almost entirely explained by the overlapping RF connectivity model. As discussed above, the variance explained by the local connectivity model was strongly correlated with both the eccentricity and polar angle models, and there was little difference between them. Second, the residual widespread effects were not biased by any imbalance in the number of data-points (nodes) across data bins. We conducted a control analysis where we equalized the number of nodes in each bin of the topographic model analysis. We used bins with 20, 30, and 50 surface nodes. For bins that contained more than the maximum number of nodes, we subsampled the data. In some subjects, a few bins contained less than the set number of nodes. We conducted the control analyses both including and excluding these bins with few nodes. Eccentricity and (lack of) polar angle residual correlations were observed in these analyses and were consistent with the original results shown in *Figure 9*, and the interpretation of the results did not change. Thus, our analyses were sensitive enough to reveal polar angle connectivity effects, and the lack of consistent residual polar angle correlations between regions with non-overlapping RFs cannot be explained by such biases in our data.

### Homotopic RF connectivity

One of the more robust effects consistently observed in resting state correlations is that of homotopic connectivity; the BOLD signals of homologous cortical regions between hemispheres are highly correlated (*Biswal et al., 1995*). Recent studies have demonstrated homotopic connectivity in visual cortex with respect to visual field representation (*Heinzle et al., 2011*; *Butt et al., 2013*). Homotopic connectivity was evident in our data when plotting inter-hemisphere correlations with respect to the radial and angular distance after reflecting the position coordinates of one hemisphere across the vertical meridian onto the other hemifield (*Figure 8—figure supplement 2*). Indeed, for inter-hemisphere comparisons, the homotopic RF predictor generally captured more

variance than the overlapping RF predictor. We tested whether homotopic connectivity (with respect to the vertical meridian) could account for the inter-hemisphere eccentricity-based connectivity effects. We removed all variance explained by the homotopic RF predictor then recomputed the residual correlations. The eccentricity effects were still present in these residual correlations (*Figure 9—figure supplement 3*), though intra-areal hV4 correlations were notably weaker. This was to be expected as the eccentricity and overlapping RF predictors were highly correlated for intra-areal hV4, and removal of the local connectivity variance also removed a large portion of the variance attributable to the eccentricity predictor. Overall, these data show that the eccentricity-based connectivity effects were not driven by homotopic RF connectivity.

## Comparison of local and widespread connectivity influences

The variance explained by the overlapping RF model (after accounting for patterns attributable to the 'noise' model) was generally greater than the variance explained by the model of residual widespread connectivity (after removal of overlapping RF connectivity effects) for within hemisphere comparisons (mean ratio across areas, conditions = 1.69:1.00). For between hemisphere comparisons, variance explained by the overlapping RF model (after accounting for patterns attributable to the 'noise' model) was weaker than the variance explained by the model of residual widespread connectivity without accounting for mirror symmetrical connections (mean ratio across areas, condition = 1.00:5.65), but was generally greater when accounting for these connections (mean ratio across areas, condition = 1.27:1.00).

In summary, these data demonstrate that both topographically local and widespread connectivity patterns significantly contributed to the observed correlation patterns across V1, V2, V3 and hV4. Critically, the eccentricity correlation patterns explained an additional, significant amount of variance in our data, which was not accounted for by the local connectivity model. For most inter-hemispheric comparisons, adding the eccentricity predictor more than doubled the variance explained by the local connectivity model, suggesting that it was the major component of the correlation patterns between hemispheres. Notably, after accounting for overlapping pRF connectivity, the strength of eccentricity correlations were similar within and across hemispheres for most comparisons between V1, V2, and V3, suggesting that this widespread connectivity pattern uniformly spans the whole visual field.

## Discussion

We investigated the spatial pattern of functional correlations across eight human early visual and extrastriate cortical areas using fMRI during conditions in which there was little or no external input (resting-state) and during conditions in which there was a dynamic external input (movie viewing). Local and widespread (spatial) patterns of correlated BOLD signal were observed in all experiments, both in the presence and absence of visual input. In agreement with prior reports, we observed topographically-local correlation patterns based on overlapping representations of visual space. In addition, we found strong evidence for topographically-widespread correlation patterns based on eccentricity within and across hemispheres, which spanned the entire visual field. The eccentricity-based correlation patterns were strongest between early visual areas (V1–V3). The effects observed in extrastriate areas (hV4, V3A–B, and VO1–2) were generally weaker than early visual areas, consistent with the coarser topographic organization of visual space and smaller surface area in these areas.

Our data extend recent findings of topographic connectivity within visual cortex using fMRI. At a fine-scale, correlation patterns reflect overlapping RFs within hemispheres and homotopic connections between hemispheres (*Heinzle et al., 2011*; *Haak et al., 2012*; *Butt et al., 2013*). By modeling V1 inter-areal connections based on overlapping RFs, Gravel and colleagues (2014) generated polar angle and eccentricity maps for V2 and V3. However, Raemaekers and colleagues (2014) reported that such fine-scaled connectivity was only observable after filtering coarse-scale components. At a coarse scale, there has been evidence for a general foveal-peripheral distinction (*Vincent et al., 2007*; *Nallasamy and Tsao, 2011*; *Smith et al., 2012*; *Raemaekers et al., 2014*), but these studies did not report a systematic *topography* of correlation patterns that reflects the underlying eccentricity organization, nor were effects of overlapping RFs and cortical distance controlled. Evidence for a more systematic relation to eccentricity was reported between V1 and ventral V3 (*Yeo et al., 2011*), though such data are also consistent with overlapping RF connectivity as well as cortical distance since only ventral V3 was probed. Here, we explicitly and quantitatively combined these connectivity phenomena in a fine-grained analysis across a wider range of brain

regions, replicating previous findings of overlapping RF and homotopic correlation patterns without filtering coarse spatial components (*Heinzle et al., 2011*; *Haak et al., 2012*; *Butt et al., 2013*; *Gravel et al., 2014*). Even after accounting for these patterns, we found a robust widespread correlation pattern that reflects the eccentricity organization across much of visual cortex. Such systematic correlation patterns suggest orderly integration processes across the whole visual field at multiple levels of the processing hierarchy, not just a foveal-peripheral dichotomy.

Eccentricity-based correlation patterns may reflect an intrinsic functional organization of visual cortex. Our results during rest demonstrate that regions with iso-eccentricity representations are likely to be co-active, even in the absence of visual input. Our results during the movie viewing condition demonstrate that the temporal dynamics yielding these eccentricity-based correlation patterns are also present during strong bottom-up input, and indicate that this organization is relevant for the processing of incoming visual input. In agreement with these findings, studies in macaques and cats have shown that the activity of neurons with similar response properties are correlated in the presence and absence of external input, suggesting that spontaneous neuronal activity is tightly linked to intrinsic cortical networks (*Arieli et al., 1995*; *Tsodyks et al., 1999*; *Goldberg et al., 2004*; *Ghuman et al., 2013*). Further, both overlapping RF and eccentricity-based connectivity patterns were observed in the presence and absence of external input, suggesting that topographically-local and widespread patterns are both part of this intrinsic functional architecture.

The observed patterns of functional connectivity may reflect both direct and indirect anatomical connectivity (*Vincent et al., 2007*; *Honey et al., 2009*). Local connectivity is likely supported by direct anatomical connections between overlapping RFs (*Cragg, 1969*; *Essen and Zeki, 1978*; *Maunsell and Van Essen, 1983*). Such wiring is necessary for the integration of information within focal points of our visual environment. Direct intra-areal anatomical connections between dorsal and ventral visual cortex at the horizontal meridians (*Jeffs et al., 2009*) and between both hemispheres at the vertical meridians (*Hubel and Wiesel, 1967*; *Essen and Zeki, 1978*; *Newsome and Allman, 1980*; *Cusick et al., 1984*; *Kennedy et al., 1986*) could support widespread functional connectivity across the visual field, though we are not aware of anatomical studies explicitly reporting eccentricity-based patterns of intra-areal connectivity. Further, while labeled cells of lateral connections in macaque striate and extrastriate cortex exhibit some anisotropy with respect to the cortical surface, this is thought to reflect cortical magnification factor, and yield isotropic visual field coverage (*Angelucci et al., 2002*). Consistent with the anatomical connectivity, we found that correlation patterns between regions with overlapping RFs were comparable along eccentricity and polar angle dimensions. Beyond overlapping RFs, correlation patterns were *anisotropic* (with respect to visual field coverage) and reflected the underlying eccentricity organization. Alternatively, the observed eccentricity-based correlation patterns may actually reflect a broader-scale anatomical organization of direct (and indirect) connections, facilitated via differences in intra-areal projections between cortical sites representing central and peripheral space (*Colby et al., 1988*; *Nakamura et al., 1993*; *Gattass et al., 2005*; *Ungerleider et al., 2008, 2014*). Such a distinction has been characterized in the patterns of supra-areal anatomical connections between early visual and extrastriate cortex in non-human primates (*Rosa, 2002*; *Gattass et al., 2005*; *Rosa and Tweedale, 2005*; *Rosa et al., 2009*; *Buckner and Yeo, 2014*). It is not known whether these anatomical connectivity patterns are 'bi-modal', and only distinguish central and peripheral space, or reflect a finer-scale organization where connectivity patterns with intermediate eccentricity representations are distinguishable from central and peripheral connectivity profiles. Our results predict that these anatomical connections should reflect a gradient, though this remains to be explored. In particular, feedback projections from extrastriate areas with receptive fields covering wide swaths of the visual field to early and intermediate visual areas could facilitate such widespread, eccentricity-dependent correlation patterns. It is interesting to note that when comparing the profile of anatomical connectivity between V2/V4 and higher order cortex (e.g., *Figure 7*, *Gattass et al., 2005*) with the organization of eccentricity across visual cortex in macaques (*Brewer et al., 2002*; *Kolster et al., 2009*; *Arcaro et al., 2011*), it is clear that higher order areas connected with peripheral parts of V2 and V4 (e.g., PO, PIP, LIP, DP, TF) have a large representation of the periphery, and higher order areas connected with foveal parts of V2 and V4 (e.g., TEO and TE) have a large representation of the fovea. The exact relationship between the observed correlation patterns and anatomical pathways will need to be further investigated.

Our data link the functional organization of early and higher order visual cortex. Previous studies have proposed eccentricity as a large-scale functional organizing principle for higher order visual cortex (*Levy et al., 2001*; *Hasson et al., 2002*; *Malach et al., 2002*). Higher order areas with foveal biases tend to be specialized in face and object recognition, and areas with peripheral biases tend to be involved in scene analysis (*Levy et al., 2001*; *Hasson et al., 2002*; *Malach et al., 2002*). Perceptually, these recognition processes require different visual acuities. For example, while fine acuity is needed for the featural discrimination among similar face and object exemplars (*Fiorentini et al., 1983*; *Goffaux et al., 2005*; *Keil, 2008*), a coarser acuity is needed for mapping the surrounding layout necessary for navigation in space (*Oliva and Schyns, 1997*; *Oliva and Torralba, 2006*). Further, eye movement patterns during scene perception are related to the types of information within a scene (*Buswell, 1935*; *Henderson and Hollingworth, 1999*). People tend to foveate on faces while orienting their peripheral vision at landscape features and room contours (*Yarbus, 1967*). These eccentricity biases are also reflected in the connectivity patterns between face and place category-selective regions in ventral temporal cortex with extrastriate visual area hV4 (*Baldassano et al., 2012*). Our data show that this divergence in the computational processes necessary for foveal and peripheral recognition is evident even in early visual cortex.

Our results underscore the importance of relating functional connectivity data to known functional (or anatomical) organization (*Jbabdi et al., 2013*; *Sporns and Honey, 2013*; *Wang et al., 2013*). The detailed retinotopic organization of visual cortex allowed for a unique opportunity to systematically compare patterns of correlated BOLD activity with the known underlying functional organization of the visual system. Across subjects and experiments, correlations were stronger between areas at matched eccentricities than within areas at large eccentricity distances, suggesting that functional connectivity analyses on BOLD data are more sensitive at revealing widespread, inter-areal connectivity patterns than the localization of individual retinotopic areas (see also *Yeo et al., 2011*). Alternative connectivity approaches not based on similarity may prove useful at revealing area boundaries (*Wig et al., 2014*), though this remains to be tested more thoroughly beyond the V1/V2 border. Thus, we propose that relating correlation patterns to known functional and anatomical data will prove important for identifying the functional pathways for the integration of information across individual, functionally specialized areas.

## Materials and methods

### Participants

14 subjects (aged 24–34 years, six females) participated in the study, which was approved by the Institutional Review Board of Princeton University (Resting State & Retinotopy Experiments: IRB#4616, Movie Viewing Experiments: IRB#5516). All participants were in good health without history of psychiatric or neurological disorders and gave their informed written consent to participate in the study and consent to publish in accordance with ethical standards set out by the Federal Policy for the Protection of Human Subjects (or 'Common Rule', U.S. Department of Health and Human Services Title 45 DFR 46). Subjects had normal or corrected-to-normal visual acuity. All participants were experienced MRI subjects that were well trained to maintain central fixation for several minutes at a time while lying still during scans.

### General procedure

All subjects participated in three scanning sessions, during which resting state scans were collected, high-resolution structural images were acquired for cortical surface reconstructions, and polar angle and eccentricity measurements were obtained to delineate retinotopic areas. 11 of these subjects viewed movie clips in a single additional scanning session.

### Resting state

Each subject participated in two versions of resting state: (1) fixation and (2) eyes closed. During the fixation scans, subjects were instructed to maintain fixation on a centrally presented dot (0.3° diameter) overlaid on a mean grey luminance screen background for 10 min. During the

eyes closed scans, the projector was turned off and subjects were instructed to keep their eyes closed for 10 min. Two runs were collected per resting condition.

## Movie condition

11 subjects viewed an audiovisual movie clip from the film *Dog Day Afternoon*. Subjects were instructed to attend to the movie, but maintain fixation on a centrally presented dot (0.3° diameter). Movie stimuli subtended 20° horizontally and 16° vertically. Two runs were collected per condition with each run lasting 5 min 45 s.

## Retinotopic mapping

Polar angle and eccentricity representations were measured using a standard traveling wave paradigm consisting of a colored checkerboard wedge or annulus, respectively (*Swisher et al., 2007*; *Arcaro et al., 2009*, *2011*). For eccentricity mapping, the annulus increased on a logarithmic scale over time in size and rate of expansion to approximately match the human cortical magnification function in early visual cortex (*Horton and Hoyt, 1991*; *Swisher et al., 2007*). Using a logarithmic scale yields a roughly even distribution of eccentricity phases across the cortical surface for early visual areas V1 and V2 (*Hansen et al., 2007*; *Swisher et al., 2007*; *Schira et al., 2009*). Stimuli mapped the central 15° of the visual field. Due to limitations of the scanner bore size and viewing angle, peripheral representations beyond 15° were not mapped nor included in any analyses. Each run consisted of eight 40 s cycles. For each subject, 4–5 polar angle runs and 2–3 eccentricity runs were collected. Early visual and extrastriate areas V1, V2, V3, hV4, V3A–B, VO1–2 were defined using standard criteria reported previously (*Sereno et al., 1995*; *DeYoe et al., 1996*; *Engel et al., 1997*; *Brewer et al., 2005*; *Wandell et al., 2007*; *Arcaro et al., 2009*). For more details, see *Arcaro et al. (2009, 2011)*.

## Data acquisition and preprocessing

Data were acquired with a 3T Skyra magnetic resonance imaging (MRI) scanner (Siemens, Munich, Germany) using a 16-channel head coil. All functional acquisitions used a gradient echo, echo planar sequence with a 64 square matrix (slice thickness of 4 mm, interleaved acquisition) leading to an in-plane resolution of $3 \times 3$ mm$^2$ (field of view [FOV], $192 \times 192$ mm$^2$; GRAPPA iPAT = 2; 32 slices per volume for resting state and 27 for movie stimuli; repetition time [TR] = 1.8 s for resting state and 1.5 s for movie scans; echo time [TE] = 30 ms; flip angle = 72°). High-resolution structural scans were acquired in each scan session for registration to surface anatomical images (MPRAGE sequence; 256 matrix; $240 \times 240$ mm$^2$ FOV; TR, 1.9 s; TE 2.1 ms; flip angle 9°, $0.9375 \times 0.9375 \times 0.9375$ mm$^3$ resolution).

## Data preprocessing

Data were analyzed using AFNI (*Cox, 1996*) (http://afni.nimh.nih.gov/afni/), SUMA (http://afni.nimh.nih.gov/afni/suma/), MATLAB (The MathWorks Inc., Natick, MA), and FreeSurfer (*Dale et al., 1999*; *Fischl et al., 1999a*) (http://surfer.nmr.mgh.harvard.edu/). Functional data were slice-time and motion corrected. Motion distance (estimated by AFNI's 3dvolreg) did not exceed 1.0 mm (relative to starting head position) in any of the six motion parameter estimates (three translation and three rotation) during any run for any subject. In preparation for correlation analyses, several additional steps were performed on the data: (1) removal of signal deviation >2.5 SDs from the mean (AFNI's 3dDespike); (2) temporal filtering retaining frequencies in the 0.01–0.1 Hz band; (3) linear and quadratic detrending; and (4) removal by linear regression of several sources of variance: (i) the six motion parameter estimates (three translation and three rotation) and their temporal derivatives, (ii) the signal from a ventricular region, and (iii) the signal from a white matter region. Removal of ventricular and white matter signal resulted in a general, broad decrease in the raw correlation values by about 0.2, though subsequent eccentricity-specific effects were slightly increased. These are standard preprocessing steps for resting-state correlation analyses (e.g., *Vincent et al., 2007*; *Yeo et al., 2011*), though our results were not dependent on these preprocessing steps, and correlation analyses on the raw data yielded qualitatively and statistically similar results. Global mean signal (GMS) removal was not included in the analysis reported here given concerns about negative correlations (*Fox et al., 2009*; *Murphy et al., 2009*; *Saad et al., 2012*), though inclusion of GMS removal yielded statistically similar results. To minimize the effect of any evoked response due to the scanner onset, the initial 21.6 s and 19.5 s were removed from each rest and movie scan,

respectively. All voxels that fell between the gray and white matter boundaries were mapped to surface model units (nodes). Only for figure illustrations from single seed correlation analyses, data were spatially filtered using a Gaussian filter to a maximum smoothness of 4 mm full-width at half-max (FWHM) (by estimating the FWHM before spatial filtering), ensuring uniformity across the surface and maintaining spatial specificity while increasing the signal-to-noise ratio (SNR) (*Chung et al., 2005*). No such spatial filtering was applied on data used for eccentricity bin or topographic model regression analyses. The timeseries from all surface nodes spanning early visual and extrastriate areas V1, V2, V3, hV4, VO1–2 (combined), V3A–B (combined), were extracted into MATLAB for correlation analyses. Eccentricity measurements were coarse for the surrounding visual cortex, so no additional extrastriate visual areas (e.g., LO1/2, TO1/2, PHC1/2, IPS0-5) were included in the analyses.

## Group seed analysis

Each subject's reconstructed cortical surface was warped to the Buckner40 template in Freesurfer (*Fischl et al., 1999b*) and then resampled in SUMA using an icosahedral shape to generate a standard mesh with a constant number of co-registered nodes (*Argall et al., 2006*). Phase and correlation maps were converted from individual surface space to the standard-mesh surface to generate group average data. Co-registered correlation maps were averaged across subjects to derive group average maps for the four seed locations. To visualize correlations as a function of visual field representation, individual subject radial and angular position data were converted to a 30 × 30 Cartesian grid space. Correlations were grouped as a function of visual field position on the grid (rounded to the nearest whole number). Multiple correlation values for the same visual position in individual subjects were averaged, and then correlation maps were averaged across subjects.

## Eccentricity bin analysis

For each subject, nodes were grouped by visual area (V1, V2, V3, hV4, V3A, V3B, VO1, VO2) for right and left hemispheres separately. Visual areas V1, V2, and V3 were separated into dorsal and ventral parts. Given the smaller surface area relative to V1, V2, and V3, visual areas V3A and V3B (as well as VO1 and VO2) were grouped together to increase the total number of samples (nodes). Due to the cortical magnification factor of early visual cortex, there were a limited number of voxels representing the periphery beyond 12.50°. Therefore, we restricted our analyses to the central 12.50°, and we refer to representations >10° eccentricity as peripheral-most. To increase signal-to-noise and control the extent of spatial signal blur, nodes were subdivided into 12 bins spanning 0.50°–12.50° eccentricity for each visual area. Eccentricity values from a log-scaled stimulus were used for the current analyses because the cortical magnification factor in early visual cortex is accounted for, yielding an approximately uniform distribution of nodes (i.e., data points) across eccentricity bins. The boundaries between bins corresponded to: 0.50°, 0.84°, 1.24°, 1.71°, 2.27°, 2.93°, 3.71°, 4.63°, 5.73°, 7.02°, 8.55°, 10.36°, and 12.50°. For each visual area, the timeseries of all nodes within each eccentricity bin were averaged to derive a mean timeseries for each eccentricity bin. In each subject, Pearson correlation coefficients were calculated between the mean timeseries of all eccentricity bins within as well as between visual areas, both within and between hemispheres. For each pair of visual areas, matrices were created containing all possible correlations between eccentricity bins. For each subject, these correlation matrices were created for each run separately and then averaged. Group average correlation matrices were also calculated for each area pair in each task (resting-fixation, resting-eyes shut, and movie-fixation). The magnitude of coefficients varied considerably between area pairs (e.g., correlation coefficients between V1 and V2 were larger than between V1 and hV4 at matched eccentricity bins). In order to illustrate the consistency in the pattern of eccentricity bin correlations across matrices (*Figures 2C, 3* only), coefficients were z-score normalized for each area pair separately. This preserved the relative differences in correlations between eccentricity bins within each matrix, but removed large magnitude differences between matrices. Non-normalized correlation matrices were used for all subsequent analyses and statistics. To ensure that the log-scaled eccentricity stimulus was not confounding the results, analyses were also run using eccentricity values that were converted into visual degrees. Comparable eccentricity-based effects were observed in this control analysis, though larger eccentricities had relatively fewer nodes per bin, and correlations with these bins were therefore more variable.

## Radial distance

Next, a ranked radial distance was calculated for each bin pair such that a radial distance of 0 corresponded to bin pairs with the same eccentricity value (iso-eccentricity) and 11 corresponded to pairs containing the foveal and peripheral-most bins. A radial distance matrix was created containing the differences between all eccentricity bin pairs (*Figure 2C*, left). To assess the relationship between the eccentricity structure and correlation patterns within the visual system, individual subject correlation matrices were correlated with this radial distance matrix.

Correlation coefficients within each matrix were then grouped as a function of radial distance. This yielded several correlation estimates for each radial distance value. Grouped correlation coefficients were then averaged to yield a single, mean correlation coefficient for each radial distance from 0–11. Correlations were Fisher z-transformed for statistical tests. For each visual area pair, two-tailed t-tests were performed on the Fisher-transformed 0–distance correlations to assess whether the subject population reliably differed from 0. False Discovery Rate (FDR) corrections were applied for each condition separately (*Benjamini and Hochberg, 1995*). A linear regression was performed across all distance correlations (0–11) for each visual area pair in each subject. For within area correlations (e.g., V1 to V1), 0 distance coefficients were excluded from the regression since the coefficients reflected correlations between identical timeseries (i.e., mean coefficients were always a value of 1). The slopes were used to evaluate the strength of eccentricity-based correlations between area pairs and across conditions. Correlation coefficients and slopes were then averaged across subjects to derive group mean distance correlations and group mean slopes for each visual area pair. Statistical significance of the group mean slopes was tested using a non-parametric permutation test in which the radial distance values were shuffled prior to grouping of individual correlations. For each iteration, the same label shuffling was applied to all subjects. Linear regression analyses were performed on each subject's permuted data and the mean slopes were calculated. This permutation was run 10,000 times. For all area pairs, the mean slopes (averaged across subjects) from the non-permuted data were larger than 97.5% of slopes from the mean permuted data (i.e., significant for a two-tailed test with α = 0.05).

As observed with the binning analysis, the overall magnitude of mean radial distance coefficients broadly varied across visual area pairs (e.g., coefficients between V1 and V2 were much larger than between V1 and hV4 at matched eccentricity differences) and across conditions. Such variability in correlation strength was orthogonal to the focus of the current study. To minimize this magnitude variability, but preserve the relation of correlations across radial distances for illustration purposes (*Figures 6, 7*), correlation coefficients were normalized to the mean correlation coefficient at 0–distance (i.e., iso-eccentricity) in individual subjects as follows:

$$d_X = 1 - (r_{0distance} - r_{Xdistance}),$$

where $r_0$ is the average correlation value at iso-eccentricity and $d_X$ is the correlation normalized to the average correlation at iso-eccentricity. This yielded a scale where 1 equals the correlation value of 0–distance. Values smaller (or larger) than 1 indicate that the correlations decrease (or increase) with greater radial distance. Since this was a simple subtraction, the relative coefficient differences between radial distances were identical to the non-normalized coefficients (i.e., preserves the slope). Importantly, the non-normalized coefficient values at and near the 0–distance were always significantly positive. Slight negative coefficients (between 0.00 and −0.15) were only observed for a few visual area pairs at large radial distance (i.e., between foveal and peripheral-most).

## Topographic model regression

The relation of local and widespread connectivity models to the observed correlation patterns was quantified across visual areas V1, V2, V3, and hh using regression. V3A–B and VO1–2 were excluded from these analyses due to the lack of published data on their population receptive fields (pRFs; *Dumoulin and Wandell, 2008*). To compare the widespread eccentricity connectivity with other models of connectivity, each hemifield map was separated into six divisions of eccentricity, each containing six divisions of polar angle, yielding a total of 36 bins for each hemifield representation. Two spatial patterns of topographically local connectivity and two patterns of topographically widespread connectivity were generated:

A. Topographically local connectivity:
1. Instrumental 'noise' connectivity (NSE): subject-specific predicted 'noise' correlations that are assumed to be non-neuronal in nature, and could result from any biases introduced from data acquisition and analyses, as well as the intrinsic spatial signal spread in the BOLD imaging (*Bandettini, 2009*). The point spread function at 3T has been estimated to be about 3.5 mm (*Engel et al., 1997*). To simulate the spatial autocorrelation in BOLD imaging at 3T, the timeseries of each voxel was replaced with a randomly generated, un-correlated timeseries for each subject's data, and a Gaussian filter with a kernel of 3.5 mm was applied to the data. These artificial data were passed through the same processing steps as the real data to derive estimated 'noise' correlations for each subject.
2. Point-to-point RF connectivity (RF): predicted correlations based on overlap of receptive fields in visual space. Due to the log-scaling used for eccentricity mapping, phase measurements were converted to visual degrees for this analysis. The receptive field of each node was calculated using a two-dimensional circular Gaussian spread (*Dumoulin and Wandell, 2008*):

$$g(x, y) = A \exp - \left[ (x - x_0)^2 + (y - y_0)^2 \right] \Big/ 2\sigma^2,$$

where $(x_0, y_0)$ is the visual field representation of a given node (in Cartesian coordinates), $A$ is normalization constant to ensure integration unity, and $\sigma$ is the Gaussian spread inferred from previously published pRF measurements (*Dumoulin and Wandell, 2008*; *Amano et al., 2009*; *Harvey and Dumoulin, 2011*; *Heinzle et al., 2011*) such that the integral of g(x, y) is 1. Specifically, a linear relationship was estimated between pRFs and eccentricity from the minimum and maximum eccentricities reported for each area. For any given area, pRF sizes varied across studies. To best approximate the pRF sizes from these prior reports, the average slope and intercept across reports was used, though analyses using individual slopes and intercepts from each of the prior reports yielded qualitatively and statistically similar results. On average, the size of the pRFs for 0.5° and 12.5° eccentricities were calculated as 0.4° and 1.6° for V1, 0.48° and 2.3° for V2, 1.0° and 4.15° for V3. For hV4, the size of the pRFs for 0.5° and 12.5° eccentricities were calculated solely from *Harvey and Dumoulin (2011)* as 1.2° and 5.8°. pRFs were constructed for each node. Individual node pRFs were binned and averaged to construct a response field for each bin. Signal spread between bins was then calculated as the amount of response field overlap between bins relative to the total response field area of the bin pair.

B. Topographically widespread connectivity:
1. Eccentricity connectivity (Ecc): predicted correlations based on radial distance. Bin pairs were assigned a value between 0 and 1; correlations were then assumed to be linearly proportional to the difference in eccentricity representations with iso-eccentricity representations assigned a value of 1.
2. Polar angle connectivity (Pol): predicted correlations based on angular distance. Bin pairs were assigned a value between 0 and 1; correlations were then assumed to be linearly proportional to the difference in polar angle representations with iso-polar angle representations assigned a value of 1.

The contribution of these four spatial patterns on the measured correlations was assessed using linear least-squares regression:

$$C(x, y) = A + \beta_1 * NSE(x, y) + \beta_2 * RF(x, y) + \beta_3 * Ecc(x, y) + \beta_4 * Pol(x, y) + \varepsilon(x, y),$$

such that for any two area pairs (x, y), the correlation pattern C(x, y) is the linear weighted sum of four modeled sources, NSE(x, y), RF(x, y), Ecc(x, y) & Pol(x, y), with separate parameter coefficients $\beta_x$, a constant $A$, and some measured error $\varepsilon$. Intra- and inter-hemisphere patterns were assessed separately. The results did not statistically differ using iterative reweighted least squares regression (bi-square), and so we only report the ordinary least squares regression results.

## Residual correlation analysis

The contribution of widespread predictors (eccentricity and polar angle) on the measured correlation patterns was re-assessed after first removing any shared variance with the topographically local connectivity. In each subject, topographically local connectivity (overlapping RF and instrumental 'noise' correlation) was removed from the data (via linear least-squares regression), and then the unexplained variance in the data (residuals) was correlated with eccentricity and polar angle predicted patterns, separately. Two-tailed t-tests were performed on the Fisher-transformed residual correlations

to assess whether the subject population reliably differed from 0. Paired t-tests were performed between eccentricity and polar angle residual correlations. False Discovery Rate (FDR) corrections were applied to each condition separately (*Benjamini and Hochberg, 1995*).

### Controlling for effects of spatial autocorrelation

Previously published pRF measurements were used to create our local connectivity model. Though these pRF size estimates were likely influenced by the point spread function of BOLD imaging, any unaccounted effect of the point spread function could lead to an underestimation of pRF coverage and thus of overlapping RF connectivity. We tested whether the observed connectivity patterns could be explained by such an underestimation of overlapping pRF effects. For each area pair, artificial data were generated with the correlation structure of the local connectivity via Cholesky decomposition. Artificial data were simulated for each run of resting and movie scans in individual subjects. These artificial data were then spatially smoothed with a 3.5 mm kernel Gaussian filter to approximate the point-spread function (*Engel et al., 1997*), and correlations were computed between all 36 bins to generate a new local connectivity model. Though the instrumental noise predictor (NSE) in the local connectivity model accounts for the same point spread function, this connectivity model extends the effects of increased spatial blur for inter-areal, overlapping RF connectivity.

### Inter-hemisphere homotopic RF connectivity

To create a model of homotopic RF connectivity (with respect to the vertical meridian), the sign of the x coordinate was flipped for the RFs in one hemisphere. Overlap was calculated in the same manner as for local overlapping RF connectivity. Topographically local connectivity and homotopic effects were removed from the data (via linear least-squares regression). The unexplained variance in the data (residuals) was then correlated with eccentricity and polar angle predictors, and statistical tests were performed on the Fisher-transformed residual correlation coefficients.

## Acknowledgements

We thank A Schapiro and L Wang for helpful comments. Supported by NSF: BCS 1025149, NIH (NINDS): F32-NS063619, NIH (NIMH): RO1-MH64043, NIH (NEI): R21-EY021078, NIH (NEI) R01-EY017699, NIH (NIMH): RO1-MH094480.

## Additional information

### Funding

| Funder | Author |
|---|---|
| National Science Foundation (NSF) | Michael J Arcaro, Ryan EB Mruczek, Sabine Kastner |
| National Institute of Mental Health (NIMH) | Michael J Arcaro, Christopher J Honey, Ryan EB Mruczek, Sabine Kastner, Uri Hasson |
| National Institute of Neurological Disorders and Stroke (NINDS) | Michael J Arcaro, Ryan EB Mruczek, Sabine Kastner |
| National Eye Institute (NEI) | Michael J Arcaro, Ryan EB Mruczek, Sabine Kastner |

The funders had no role in study design, data collection and interpretation, or the decision to submit the work for publication.

### Author contributions

MJA, CJH, REBM, Conception and design, Acquisition of data, Analysis and interpretation of data, Drafting or revising the article; SK, UH, Conception and design, Analysis and interpretation of data, Drafting or revising the article

### Ethics

Human subjects: This study was approved by the Institutional Review Board of Princeton University (Resting State & Retinotopy Experiments: IRB#4616, Movie Viewing Experiments: IRB#5516). All participants were in good health without history of psychiatric or neurological disorders and gave

their informed written consent to participate in the study and consent to publish in accordance with ethical standards set out by the Federal Policy for the Protection of Human Subjects (or 'Common Rule', U.S. Department of Health and Human Services Title 45 DFR 46).

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
