## [Decision Letter]

Thank you for sending your work entitled “Widespread correlation patterns of fMRI signal across visual cortex reflect eccentricity organization” for consideration at *eLife*. Your article has been favorably evaluated by Eve Marder (Senior editor), Timothy Behrens (Reviewing editor), and 3 reviewers.

The Reviewing editor and the reviewers discussed their comments before we reached this decision, and the Reviewing editor has assembled the following comments to help you prepare a revised submission.

You can see the comments below. During our discussion, the central concerns with your manuscript are the overlap with Raemaekers et al. Neuroimage 2013 and disagreement between the two studies of the extent of the bias towards eccentricity rather than homotopic or polar connectivity.

It will be very important in your revisions to include a thorough discussion of these points. It would also be useful if there are further analyses you can try that may address these issues directly. It is of particular interest whether your data quality are adequate to disambiguate the different possibilities (see last comment R3). It was clear from the discussion among the reviewers that it is essential to address this point if *eLife* are to consider publishing a revision.

*Reviewer #1*:

This is an impressively careful and thorough study of an important set of phenomena related to resting-state functional connectivity in human visual cortex. Several previous studies cited by the authors have identified resting-state correlations that include retinotopic components and other components that are spatially more widespread. The current study systematically examines the spatial pattern of correlations using two resting-state paradigms (with eyes closed and eyes open), plus a paradigm involving watching a movie while fixating. The most significant finding is that for all 3 paradigms the spatially extended correlations are stronger along iso-eccentricity bands. There is also a weaker component related to polar angle and another component related to homotopic (mirror-image retinotopy) in the opposite hemisphere.

In the Discussion, the authors attempt to provide a neurobiologically plausible interpretation of the strong eccentricity-related functional correlations analyzed in this study. In this reviewer's opinion, the results remain intriguing but still quite puzzling. However, the careful analysis done in this study represents a major advance, and it will likely stimulate further explorations aimed at better understanding the functional significance of these phenomena.

The manuscript is well reasoned and well written, and needs only minor revisions in the opinion of this reviewer.

Minor comments:

1) Both the Abstract and Introduction refer to evidence for more than two dozen retinotopic areas in human visual cortex. However, neither of the cited articles (78; 89) actually document evidence for two dozen retinotopic areas. Either the statement should be toned down (it is not critical to the study) or appropriate references should be added.

2) Results section, Materials and Methods section and Figure 2.

Please state the extent of visual stimuli for the retinotopic mapping (was it 30 degrees along the meridia, as in Arcaro et al., 2011?). Also, add a scale bar for the eccentricity maps in Figure 2. Finally, consider clarifying that 'peripheral' really means 'near periphery', since nearly half of retinotopic cortex representing the far periphery (> 11 deg) was not mapped.

3) Results section, paragraph 12, and Figure 7 legend. The legend states that “Average individual subject inter-run correlations between hemispheres......steadily decreased at larger distances for all conditions”, which doesn't match what the figure actually shows. The text has it right: “Indeed, inter-run Fisher-transformed correlations on resting state data showed no effect of radial distance (all ps > 0.05), validating the assumption that noise correlations should not be reliable across runs (Figure 7, blue and black lines).” Please rectify the confusing statement in the legend.

*Reviewer #2*:

Overall I think this work presents a carefully conducted study. The level of careful evaluation and discussion is very prominent and this work should find an interested readership within the vision neuroscience community.

The authors use a seed-based approach to investigate the spatial distribution of connectivity patterns in rs-FRMI data. Interestingly, this was done using both eyes-closed and 'fixation' resting state fMRI. The paper further suggests that the retinotopic connectivity patterns (during rest as well as movie viewing) are largely organized along iso-eccentricity bands, which might reflect an important feature of cortical visual processing.

The findings are presented along with a number of important checks to rule out some of the more obvious confounds, such as BOLD smearing, eye-movements, subject motion and other nuisances.

The overall bottom-line finding (as e.g. revealed by the authors choice of title and impact statement) of correlation patterns reflecting eccentricity organisation in principle has been demonstrated multiple times before (see e.g. discussion of foveal vs high eccentricity representations in [79] PNAS or [46] Neuroimage). Particularly the latter paper already demonstrates that the retinotopic organization of visual cortex can be probed using resting state connectivity fMRI.

*Reviewer #3*:

This manuscript describes a study into the functional connectivity across several human visual cortical areas derived from conventional fMRI resting-state measurements (under fixation and eyes-closed conditions) and measurements while subjects viewed a movie stimulus. The main finding is that the connectivity exhibits a striking organization across the visual cortical areas in which locations that retinotopically represent the same eccentricity of the visual field tend to be correlated, across adjacent cortical areas as well as across the ventral and dorsal subdivisions and across hemispheres. The consistency of this organization throughout the visual cortical areas argues that it is present both in the presence and absence of bottom-up anatomical connectivity. Steps are taken to control for effects such as receptive field size and spatial spread based on modeling their effects using previously reported measurements.

Overall this manuscript is extremely well written and the results are compelling. Methodologically the approach is well thought out and carefully executed. The intra-run correlation of the movie watching data presented in Figure 7 and the dorsal-ventral correlations in Figure 4 were particularly compelling.

*My main concern is in the interpretation of these striking findings. As the authors mention, functional connectivity reflects both direct and indirect anatomical connections, and the authors also mention that there have been no reports of anatomical connectivity that could support the observed correlations across eccentricities. The foveal-peripheral biases that are reviewed could be consistent with these observations. While I acknowledge that this is beyond the scope of the current report to identify the origin of this organizational feature, perhaps the manuscript could address further this issue which may be very challenging to reconcile*.

The issue of the effective resolution along the eccentricity coordinate versus the polar angle coordinate was not discussed, and I wonder how this might affect the findings. Because the cortical areas are elongated along the eccentricity direction there are more voxels to sample the progression of eccentricity than there are to sample the progressions of polar angles. Upon examining the plots in Figure 2 (and to some extent the plots in Figure 3) it appears that there is some consistent correlations across all areas averaged in the retinotopically corresponding region (especially well seen in seeds centered on 1 deg, 2.5 deg, and 5 deg), but also there seems to be some symmetry in the correlation patterns across quadrants in the 2.5 deg and 5 deg maps. One wonders if it is possible that there is a polar angle component to the organization as well, i.e., the correlations may not be only a function of eccentricity but could be a function of polar angle as well. This question gets at my main concern outlined above. Given the reduced resolution in the polar angle direction this question may be difficult to address using this data, and some attempt is made in Figure 8, but if the resolution is not sufficient perhaps the authors could discuss the potential of an unresolved polar angle organization as an alternate possibility.

**[Editors' note: further revisions were requested prior to acceptance, as described below.]**

Thank you for resubmitting your work entitled “Widespread correlation patterns of fMRI signal across visual cortex reflect eccentricity organization” for further consideration at eLife. Your revised article has been favorably evaluated by Eve Marder (Senior editor), by Timothy Behrens (BRE), and by the original 3 reviewers.

Often the BRE will choose to handle revisions himself, but on this occasion Tim Behrens wanted to return to the referees as there were some detailed points in the revisions that required their analytical expertise.

We are now almost ready to accept for publication, but there are a few remaining issues about presentation.

*The main issue is that one reviewer is still concerned by the absence of any reflection of polar geometry in the data, and by the apparent new claim that this cannot be due to the quality of your data. During the course of the revision, a second paper with higher quality data has shown that it is possible to see reflections of polar geometry in resting connectivity (Gravel et al., http://journal.frontiersin.org/Journal/10.3389/fnins.2014.00339/)*.

The review panel are not sure why this is difficult to find in your data, but we would like you to be clear in the Introduction and the discussion that such structure is visible in higher quality data. ”

In concert with these changes, we would like you to be slightly clearer about the novelty of your claim with respect to other papers that have shown eccentricity organisation’s – albeit with less detail and precision than you show here.

For example, we do not think that the following statement in the Introduction is a fair reflection of the literature:

“In addition, widespread functional correlation patterns have been observed across visual cortex in both macaques (57; 87) and humans (65; 66; 94; 27). The governing principles, if any, of such widespread connectivity patterns are still unknown.”

Reviewer 3 had related concerns that I reproduce here. We do not require you to perform the analysis in point 1 below, if you are clear (as above) that the polar connectivity can be seen in higher quality data, but if you choose to do the analysis, that would be welcome. However, we would like you to address point 2.

When you resubmit your manuscript, it will not go back to review, but will be assessed by BRE member Tim Behrens.

*Reviewer #3*:

I thank the authors for responding my concerns and for adding the analyses shown in Figures 12 and 13. The new results presented in Figure 12, showing that in some cases within and across V2 and V3 (although V1 is not included?) the correlation falls with angular distance, are particularly helpful.

*I am not sure, however, that the concern I had raised about the potential asymmetry in the resolving power along the eccentricity coordinate and polar angle coordinate has been completely addressed. First, the analyses are restricted to only eccentricities from 1 degree to about 12 degrees, so it is not clear how much a given fMRI voxel will sample along the eccentricity direction relative to the polar angle direction given the cortical magnification and therefore how marginal the effect would be. Also, the new analysis accounting for the potential statistical biases arising from re-binning the data does not address the fact that there is higher effective spatial resolution of the eccentricity coordinate than the polar angle coordinate. The plots in Figure 12 are the most direct measure of this, since the observation that in some regions the correlation falls off with distance argues that there may be in some cases enough voxels along the polar angle direction to properly resolve a spatial gradient of correlation. Perhaps the authors could address this issue more directly. (Smoothing the data along the eccentricity direction to yield similar resolution for both directions may be one brute-force approach, but perhaps there is another, better way.*)

*Still, the new results provided in Figure 12 highlight that there is some level of “polar angle connectivity”, which was previously only indirectly demonstrated, e.g., in the second and third panels of Figure 2 and in Figure 8. As the authors point out, based on the topographic model regression analyses there is evidence for “local connectivity patterns” including overlapping RF effects and correlations between homotopic regions. Beneath these overlapping RF and homotopic correlations there are also the widespread correlations, and this pattern appears to be weaker in the polar angle direction. Perhaps the authors would consider including the results of Figure 12 in the main manuscript, and boiling down the results of Figure 9 by reporting the overall variance explained by local connectivity versus widespread connectivity and what proportion of the widespread connectivity is along the eccentricity direction and what proportion is along the polar angle direction. While the authors do point out that these four spatial components are not orthogonal, some clarification of how much of the observed correlations can be explained by widespread patterns would strengthen the manuscript*.

---

## [Author Response]

Reviewer #1:

Below, we address each of the reviewer’s specific points.

*Minor comments*:

*1) Both the Abstract and Introduction refer to evidence for more than two dozen retinotopic areas in human visual cortex. However, neither of the cited articles (78; 89) actually document evidence for two dozen retinotopic areas. Either the statement should be toned down (it is not critical to the study) or appropriate references should be added*.

We agree with the reviewer that these two articles do not discuss the organization of each area in detail; however, these reviews serve as a general reference for the extent of retinotopic areas across the visual system. These two review articles reference a combined 24 cortical retinotopic areas: V1-3, hV4, VO1-2, PHC1-2, V3A-B, V6, LO1-2, TO1-2, IPS0-5, SPL1, PreCC/SFS, PreCC/IFS and 2 sub-cortical areas: LGN and superior colliculus. References to the particular mapping papers discussing the organization of each of these areas can be found in these review articles. We initially had referenced the original mapping studies. To appropriately cite articles that document the topographic organization in detail for all of these areas, we need to cite a minimum of 12 papers. Due to space limitations of the introduction section, we were not able to include all references to these original papers, and we were only able to cite the two reviews. To address the reviewer’s concern, we added two additional review paper references and explicitly note that interested readers should use theses review papers as a resource, which can refer readers to additional references and the original mapping studies:

“Through the use of functional magnetic resonance imaging (fMRI), it has now become evident that the human visual system contains over two-dozen visual maps (for review and references to original mapping studies, see [88], [78], [89], [1]).” paragraph 1 in Introduction section.

*2) Results section, Materials and Methods section and Figure 2*.

*Please state the extent of visual stimuli for the retinotopic mapping (was it 30 degrees along the meridia, as in Arcaro et al., 2011?). Also, add a scale bar for the eccentricity maps in Figure 2. Finally, consider clarifying that 'peripheral' really means 'near periphery', since nearly half of retinotopic cortex representing the far periphery (> 11 deg) was not mapped*.

The extent was indeed 30°. We have added this to the text in the Results section and Materials and Methods section.

We have added outlines of 1.8°, 5.5°, and 15°. Since the eccentricity phases were derived from log-scaled stimuli, distance from the fovea in visual degrees is nonlinear. As discussed in the methods section, the log-scaled mapping stimulus accounts for the cortical magnification factor and yields an approximately uniform distribution of data points across eccentricities. These outlines should provide a good reference for correspondence to visual field position.

We agree with the reviewer that we did not evaluate the ‘far periphery’ in our binning analyses. Our eccentricity mapping covered the central 15°. By conventional standards, we agree that this is not considered the ‘periphery.’ In the text, we now specifically state the eccentricity ranges that we consider ‘peripheral’ in the Results section and Materials and Methods section. We also state that we did not map responses in the far-periphery (i.e. above 15°), and changed references of periphery to ‘peripheral-most’.

As discussed in our manuscript, the number of voxels representing 12.5° – 15° eccentricities was minimal due to cortical magnification. As such, we restricted our binning analyses to the central 12.5°, not the central 11° as stated by the reviewer. For the eccentricity binning analysis, we grouped the data into 11 bins. We assume that this confusion arose from Figures 6 and 7 where we labeled the bins on the x-axis as ‘Radial Distance’, which could be inferred as visual degrees. We have changed the x-axis label to read ‘Radial Bin Distance’ and clarified the x-axis scale in the figure legend.

*3) Results section, paragraph 12, and Figure 7 legend. The legend states that “Average individual subject inter-run correlations between hemispheres......steadily decreased at larger distances for all conditions”, which doesn't match what the figure actually shows. The text has it right: “Indeed, inter-run Fisher-transformed correlations on resting state data showed no effect of radial distance (all ps > 0.05), validating the assumption that noise correlations should not be reliable across runs (Figure 7, blue and black lines).” Please rectify the confusing statement in the legend*.

We thank the reviewer for catching this error. We have corrected the legend to state:

“For the movie viewing condition, average individual subject inter-run correlations between hemispheres as well as between dorsal and ventral portions of V2 and V3 were strongest at the 0-radial distance (iso-eccentricity) and steadily decreased at larger distances. For resting data, correlations did not vary as a function of radial distance. See Figure 6.”

*Reviewer #2*:

*The overall bottom-line finding (as e.g. revealed by the authors choice of title and impact statement) of correlation patterns reflecting eccentricity organisation in principle has been demonstrated multiple times before (see e.g. discussion of foveal vs high eccentricity representations in [79] PNAS or [46] Neuroimage). Particularly the latter paper already demonstrates that the retinotopic organization of visual cortex can be probed using resting state connectivity fMRI*.

We agree with the reviewer that a few prior papers have demonstrated a coarse distinction in the patterns of activity during ‘rest’ across cortex representing ‘foveal’ and ‘peripheral’ visual space ([79], ; [87]; see also ; [63]). We briefly reviewed this in our Discussion section. In contrast to our study, however, these prior studies do not show a large-scale, topographic correlation pattern that reflects the underlying fine-scale eccentricity organization. We have added text to the Discussion to address some of these concerns. Here, we further discuss why our current study provides a substantial new understanding of visual cortical correlation patterns beyond what previously has been shown:

1) We demonstrated a topography of correlation patterns that reflects the underlying, fine-scale eccentricity organization. Prior studies found a coarse distinction between ‘foveal’ and ‘peripheral’ cortex using data-driven analyses (e.g., ICA, clustering), but did not look for / or report a systematic organization reflecting the underlying eccentricity map. As we discuss in our study, coarse center-periphery distinction not only has been found in early visual areas, but also in higher order extrastriate areas ([58]. ; [45]; [59]). Its origin may reflect two very different gross processes computed at foveal and peripheral areas (Malach, 2004 TiCS). In contrast, our findings report a novel fine-scale topography of eccentricity-based correlation patterns, in which, for example, local regions representing 2°, 5°, 10° eccentricities are most correlated, respectively, with other regions representing 2°, 5°, 10° eccentricities within and across early visual (V1-V3) and extrastriate (hV4) areas. Such a systematic correlation pattern suggests orderly integration processes across the whole visual field at multiple levels of the processing hierarchy, not just a foveal-peripheral dichotomy. As such, we feel that these eccentricity-specific correlation patterns go substantially beyond prior work.

2) As the reviewer stated, Heinzle and colleagues demonstrated that the retinotopic organization of visual cortex, specifically between areas V1 – V3, can be probed using resting state connectivity measures on fMRI data. Consistent with Heinzle and colleagues, we found retinotopically-specific correlation patterns based on overlapping response fields between V1 and V3. Importantly, however, we also found significant overlapping response field effects for other area pairs including V2 and hV4. Distinct from the Heinzle study, we found eccentricity-specific correlation patterns that cannot be accounted for by overlapping response fields. Heinzle and colleagues did not probe such separate eccentricity-based connectivity. As such, we feel that our study provides new insight into the intrinsic organization of visual cortex beyond the Heinzle study while also validating this previous report of overlapping receptive field connectivity. We have already cited the Heinzle study in our Discussion about overlapping receptive field connectivity and have discussed extensively the novel aspects of our study.

*Most importantly, Raemaekers et al (2014, Neuroimage) present a detailed analysis of ICA-derived spatial networks in V1-3 from high-resolution rs-FMRI at 7T. These authors argue for more complicated overlapping organisations (eccentricity as well as polar angle based), where the finer-grained organisation emerges only after filtering large-scale network fluctuations. This leads to a more extensive set of findings in Raemaekers et al. relative to this work. A discussion of their paper would be crucially important but is lacking from this manuscript. This also needs to include a thorough discussion of methodological differences, e.g. the use of multi-variate rather than uni-variate statistical methods. It appears that the inherent limitations of seed-based analysis (in terms of difficulties accounting for additional signals) may be responsible for differences between the two manuscripts*.

We thank the reviewer for raising these important points.

In relation to the first point (relation between the present study and that of [70]), we have now expanded our Discussion of the [70] paper, as suggested, and have made clearer how the present results relate to and extend that prior work:

“At a fine-scale, there have been correlation patterns reflecting overlapping RF within hemispheres and homotopic connections between hemispheres (46; 42; 18; 70). Raemaekers and colleagues (2014) recently reported that such fine-scaled connectivity was only observable after filtering coarse-scale components. Consistent with prior studies (46; 42; 18), we observed overlapping RF within hemispheres and homotopic connections between hemispheres without such data filtering. At a coarse scale, there has been evidence for a general foveal-peripheral distinction (87; 63; 79; 70), but these studies did not report a systematic topography of correlation patterns that reflects the underlying eccentricity organization, nor were effects of overlapping RFs and cortical distance controlled. Evidence for a more systematic relation to eccentricity was reported between V1 and ventral V3 (94), though such data are also consistent with overlapping RF connectivity as well as cortical distance since only ventral V3 was probed. Here, we explicitly and quantitatively combine all of these connectivity phenomena in a fine-grained analysis and across a wider range of brain regions, replicating previous findings of overlapping RF and homotopic correlation patterns.”

In relation to the second point (methodological differences and univariate vs. multivariate analyses), we present arguments and analyses that support our methodological approach, and buttress the novelty of our findings.

To summarize the prior relevant findings: Raemaekers and colleagues (2014) demonstrated 1) an effect of overlapping response field connectivity between visual areas V1, V2, and V3 with correlations decreasing between areas as a function of visual degree distance, 2) homotopic correlations across the right and left hemispheres within individual areas, and 3) coarse ICA components spanning various portions of visual cortex. Importantly, the study by Raemaekers and colleagues did not report fine scale eccentricity-based correlation patterns. As stated above, we have expanded our original discussion of this paper in our Discussion section. Here, we further discuss the differences between our study and the Raemaekers and colleagues (2014) study:

Overlapping response field connectivity:

1) Similar to our results and the Heinzle study, Raemaekers and colleagues found significant functional connectivity effects based on overlapping response fields. As opposed to the Heinzle paper and our paper, Raemaekers and colleagues only observed these effects after filtering their data with ICA components. Consistent with Heinzle and colleagues, we found significant overlapping response field connectivity effects without such filtering. Raemaekers and colleagues discussed this difference in regards to the Heinzle study (see [70], p. 918-9). In their discussion of the Heinzle study, Raemaekers and colleagues noted a few differences in the methods and analyses, but ultimately concluded: “Whether this difference in methods can fully explain the difference in results is unclear…” We agree that there was considerable variability in the imaging and analytical approaches across all three studies (Heinzle’s, Raemaekers’, and ours). As such, it is unclear why filtering was needed in the Raemaekers study to observe topographic patterns of connectivity, but not in our study or the Heinzle study.

Eccentricity-specific connectivity:

1) We observed eccentricity-specific topographic correlation patterns distinguishable from overlapping response field connectivity. Raemaekers and colleagues did not find a significant difference in their topographic connectivity analyses between radial or tangential dimensions during ‘rest’ (see Figure 4 in [70]), but they also did not directly test for eccentricity-specific connectivity distinct from overlapping response fields (e.g., between dorsal and ventral cortex). Raemaekers and colleagues discuss the lack of a radial/tangential distinction in their discussion: “The results regarding a radial/tangential connectivity bias were inconclusive” (see [70], p. 919). Given that we found strong evidence for eccentricity-specific topographic connectivity across several analyses, we feel that our current study provides novel results not previously shown (or directly tested) by Raemaekers and colleagues.

2) Raemaekers and colleagues evaluated polar angle and eccentricity distance connectivity effects during ‘rest’ only for intra-areal patterns (see Figure 6 in [70]). We found that intra-areal correlation patterns differ from inter-areal correlation patterns, and likely contain strong anatomical distance effects (see our Figure 6—figure supplement 2). We found that eccentricity-specific correlation patterns were most apparent between regions where (anatomical) distance and overlapping response field effects were minimal (e.g., between dorsal and ventral V2 / V3). This was not directly evaluated / reported in the Raemaekers study.

3) Raemaekers and colleagues observed significant topographic connectivity effects for polar angle and eccentricity mapping experiments (see Figure 3 in [70]). Such topographic patterns were likely to be induced by the correlation in the stimulus structure, and does not necessarily reflect intrinsic polar angle or eccentricity connectivity (i.e., connectivity was driven by the stimulus systematically moving across visual space along either radial or tangential dimensions).

4) Raemaekers and colleagues reported general ‘foveal’ and ‘peripheral’ ICA components in 5 subjects (see Figure 7 in [70]). These ICA components appear consistent with previously reported foveal and peripheral distinctions (see above section discussing [79]). As discussed above, this differs from our results, which demonstrates fine-scale, topographic correlation patterns that reflect the underlying eccentricity organization. In the Raemaekers’ study, the ICA components were not directly compared with eccentricity maps. In fact, the yellow peaks of the peripheral ICA components (see their Figure 7) fall outside their retinotopically-identified visual areas. Further, there was no evaluation of the consistency of these ICA components across subjects and additional ICA components orthogonal to the ‘foveal’ and ‘peripheral’ ICAs were also reported. As such, it is difficult to interpret the significance and consistency of these ICA components, and their specific relation to the retinotopic organization of visual cortex. Below in this letter, we further investigate the interpretability of ICA components and their correspondence to the effects we report in the present manuscript.

Methodological differences:

Numerous methodological differences between our study and Raemaekers and colleague’s study may account for the increased sensitivity in our study to the large-scale eccentricity organization.

1) There were significant differences in the extent of the visual field mapped. We mapped eccentricity representations out to 15° (from central fixation) and included the central 12.5° in all of our bin analyses. Raemaekers and colleagues only mapped out to 7.5°. Though we observed significant effects of eccentricity even within a few degrees of visual space (see Figure 6), the restricted coverage of visual space may have limited Raemaekers and colleagues’ ability to reliably detect eccentricity-specific correlation patterns.

2) The filtering applied by Raemaekers may have removed key components. In their discussion, Raemaekers and colleagues acknowledged that the data filtering might have removed a foveal / peripheral distinction in their topographic connectivity analyses (see [70], p. 919): “There were, however, complicating factors that may have caused a lack of significant results. Some of the coarse-scale networks that were detected with ICA contained features that would either contribute to a radial or tangential connectivity bias, and this may have obscured results in the initial analysis… The fluctuations within these (radial/tangential) networks were, however, filtered from the data in the second analysis, which may have attenuated the observed effect.”

3) As the reviewer suggested, it is possible that the difference between the Raemaekers study and our study reflects differences between data-driven multivariate and seed-based correlation analyses. Multivariate methods are potentially more powerful than univariate approaches, but their output can also be more difficult to definitively interpret. This is the case for complex methods such as ICA, and especially for spatial ICA applied to fMRI data, where the basic BOLD signal properties that separate components remain under debate (Daubechies, PNAS 2009). Prior studies using ICA analyses have only identified a broad foveal-peripheral distinction in ‘rest’ data. Here, we demonstrate topographic correlation patterns across the central 15° of visual space that reflect the underlying eccentricity organization.

It is possible that ICA analyses may be less prone to revealing such topography. To investigate the ability of ICA analyses to reveal fine-scaled eccentricity organization, we applied ICA (FSL’s MELODIC) to data from an eccentricity mapping experiment (Figure 10).Author response image 1.Spatially-specific ICA components from stimulus-driven (eccentricity mapping) data.

We used eccentricity-mapping data, since the ring stimulus evokes fine-scale eccentricity-specific topographic patterns of activity that can be differentiated in the BOLD signal. As seen in the phase maps, the underlying structure of activity patterns is spatially contiguous, not discrete, and can be clearly identified with conventional retinotopic mapping analyses (see [29]; [82]; [6]). In each subject, we identified about 45 ICA components. Most components were associated with distributed noise or relegated to non-visual networks (e.g., DMN and fronto-parietal). Across 5 subjects, we consistently identified between 2-3 ICA components in these data that were localized to contiguous parts of visual cortex and corresponded to spatially specific representations of the visual field. Each ICA component spanned several degrees of eccentricity. The ‘Peripheral’ ICA component was located largely outside of our eccentricity maps, and likely corresponds to regions of visual space >15° eccentricity. The ‘Mid’ ICA component spanned a region of visual space ranging between 4° – 8° eccentricities. In 3 of 5 subjects, we identified a ‘Foveal’ ICA component around the occipital pole and lateral surface that corresponded to foveal space less than ∼2° eccentricity.

In one subject, we identified another ICA component overlapping 13 - 15° eccentricity and part of ICA 1. ICA components did not reveal any finer-grained organization such as seen in the phase maps. In a few subjects, there were 1-2 additional ICAs that did not appear to be spatially-specific, and either spanned most of the visual field or covered multiple, non-contiguous eccentricity ranges (e.g., covered both ∼1° - 4° and ∼8° - 10°). Though these results demonstrate coarse eccentricity organization and could be consistent with the existence of a topographic organization, the components by themselves do not reflect a fine-scaled eccentricity organization as shown in the phase maps. These data suggest that such ICA analyses may not be sensitive enough to reveal fine-scale spatial gradients in data (also see Daubechies et al. 2009 PNAS), and may demonstrate why previous studies employing ICA analyses only report broad foveal-peripheral distinctions.

Upon close inspection of our ICA analyses with prior data, it appears that the previously reported ‘peripheral’ ICA in resting state studies does not correspond to the range of eccentricities probed in our current experiment. In our ICA analyses, our ‘Peripheral’ ICA component was located in anterior parts of the calcarine sulcus and retrosplenial cortex. This anatomically corresponds well with the ‘peripheral’ ICA previously reported by [79] and [70] (see Figure 10, right column). However, this component falls mainly outside of the retinotopically defined visual areas (> 15° eccentricity) in our current study (see Figure 10, white line on flat maps). The ‘Mid’ and ‘Foveal’ ICA components were located around the occipital pole and overlap with each subject’s eccentricity map. These ICAs anatomically appear to correspond to the ‘foveal’ component previously reported. As such, the prior ‘foveal’ and ‘peripheral’ differentiation in resting state data may not be directly relatable to the eccentricity-based correlation patterns shown in our current study.

4) Though our ICA analysis distinguished foveal from peripheral cortex, it was not able to reveal a fine-scaled topography from eccentricity mapping data. Next, we tested whether other data-driven analyses based on correlation patterns between individual data points (i.e., nodes) could reveal such organization. We used a k-means clustering algorithm to segment our data. At a coarse segmentation (k=2), we observed a parcellation of visual cortex in individual subjects that reflects a foveal / peripheral distinction (note: this distinction is within our retinotopic maps, not outside like the ICA analyses). At larger k-segmentations, additional clusters appeared to be symmetric around the fovea, consistent with an eccentricity-based organization. At larger k-segmentations, there was also some differentiation between early visual (V1-V3) and higher-order cortical areas (V3A/B, LO / VO / MT / IPS), which is consistent with observations from our binning analyses and prior reports (79). i.e., as seen in our binning analyses, peripheral regions of V1 were more strongly correlated with peripheral regions of V3A/B (vs. foveal V3A/B) even though peripheral regions of V1 were more strongly correlated with peripheral V3 (vs. peripheral of V3A/B). This differentiation between early visual and higher order areas is orthogonal to the current question of eccentricity-based connectivity patterns. To illustrate the relation of these clusters to the underlying eccentricity organization while minimizing the broad regional differentiation between early visual and higher-order cortex, we color coded each cluster based on the strength of their correlation with the foveal cluster in early visual cortex. In this visualization, a cluster organization similar to the eccentricity organization is apparent for segmentations based on intra- and inter- hemisphere correlations. See Figure 11.Author response image 2.K-Means Cluster Analysis.

This analysis demonstrates that a gradient of eccentricity-based correlation patterns similar to that revealed in our seed-based analyses can be identified with data-driven segmentation methods. However, clustering was variable across subjects, and multiple organization patterns were evident in the data. As such, we feel that our current correlation analyses are better suited for examining functional connectivity patterns within visual cortex specifically related to eccentricity organization.

Reviewer #3:

*My main concern is in the interpretation of these striking findings. As the authors mention, functional connectivity reflects both direct and indirect anatomical connections, and the authors also mention that there have been no reports of anatomical connectivity that could support the observed correlations across eccentricities. The foveal-peripheral biases that are reviewed could be consistent with these observations. While I acknowledge that this is beyond the scope of the current report to identify the origin of this organizational feature, perhaps the manuscript could address further this issue which may be very challenging to reconcile*.

We agree with the reviewer that the structure-function relationship of the eccentricity-specific correlation pattern needs to be explored and deserves further discussion. A foveal-peripheral distinction in connectivity has been noted in several prior anatomical studies (20; 62; 36; 84; 85) and does offer a plausible anatomical basis for the functional eccentricity-based correlation patterns observed in our data. We have expanded discussion of a possible structure-function relationship:

“Such a distinction has been characterized in the patterns of supra-areal anatomical connections between early visual and extrastriate cortex in non-human primates (71; 36; 73; 72; 16). It is not known whether these anatomical connectivity patterns are ‘bi-modal’, and only distinguish central and peripheral space, or are part of a topography where connectivity patterns with intermediate eccentricity representations are distinguishable from central and peripheral connectivity profiles. Our results would predict that the anatomical connections reflect a gradient, though this remains to be explored. In particular, feedback projections from extrastriate areas with receptive fields covering wide swaths of the visual field to early and intermediate visual areas could facilitate such widespread, eccentricity-dependent correlation patterns. It is interesting to note that when comparing the profile of anatomical connectivity between V2 / V4 and higher order cortex (Figure 7; [36]) with the organization of eccentricity across visual cortex ([15]; [56]; [7], also see ; [6] and ; Kolster et al. 2010), it is clear that higher order areas connected with peripheral parts of V2 and V4 (e.g., PO, PIP, LIP, DP, TF) have a peripheral visual field bias and higher order areas connected with foveal parts of V2 and V4 (e.g., TEO, TE) have a foveal visual field bias.”

*The issue of the effective resolution along the eccentricity coordinate versus the polar angle coordinate was not discussed, and I wonder how this might affect the findings. Because the cortical areas are elongated along the eccentricity direction there are more voxels to sample the progression of eccentricity than there are to sample the progressions of polar angles. Upon examining the plots in*
Figure 2
*(and to some extent the plots in*
Figure 3*) it appears that there is some consistent correlations across all areas averaged in the retinotopically corresponding region (especially well seen in seeds centered on 1 deg, 2.5 deg, and 5 deg), but also there seems to be some symmetry in the correlation patterns across quadrants in the 2.5 deg and 5 deg maps. One wonders if it is possible that there is a polar angle component to the organization as well, i.e., the correlations may not be only a function of eccentricity but could be a function of polar angle as well. This question gets at my main concern outlined above. Given the reduced resolution in the polar angle direction this question may be difficult to address using this data, and some attempt is made in*
Figure 8*, but if the resolution is not sufficient perhaps the authors could discuss the potential of an unresolved polar angle organization as an alternate possibility*.

We thank the reviewer for raising this point. We agree that it is very important to consider the effective resolution along eccentricity and polar angle dimensions. An imbalance of voxels between eccentricity and polar angle dimensions could affect interpretation of the results, in particular, whether any such polar angle connectivity is detectable at our current imaging resolution. This could manifest it two ways: 1) if the sampling space of individual voxels is greater along the eccentricity dimension than the polar angle dimension, and 2) if the distribution of voxels across bins in our topographic model analysis is imbalanced. When analyzing the data, we were well aware of these potential issues, and feel that aspects of our original analyses suggest that this was not actually a major issue. In addition, we performed a few new control analyses to further rule out this potential issue. Though our control analyses provide strong evidence for a lack of polar angle connectivity, we agree that this is an important point that deserves mention in the current study, and now include a description of the issue along with a new control analysis in paragraphs 16-18 of the Results section.

1) As the reviewer noted, cortical areas V1, V2, and V3 are elongated along the eccentricity dimension. As such, a given voxel will generally sample a greater extent along the polar angle dimension than eccentricity, though this becomes marginal as one moves further into the periphery due to the cortical magnification factor. This imbalance in sampling at the individual voxel level could have limited our ability to detect polar angle-based connectivity patterns. In the topographic model analysis, we did not find a consistent effect of polar angle connectivity after accounting for overlapping response fields (see Figure 9). However, we found that eccentricity and polar angle models equally contributed to the variance explained by the overlapping RF model paragraphs 18-19 of Results section. That is, we found some effect of polar angle connectivity, but it could be attributed to overlapping RFs. Given this dissociation in polar angle connectivity patterns with respect to within and outside the extent of an overlapping RF, we feel that our imaging resolution and analytical approach was sensitive enough for detecting polar angle-based connectivity.

2) To further evaluate whether an effect of polar angle-based connectivity was detectable in our data, we performed the binning analysis with respect to angular distance (collapsing across eccentricities). See Figure 12. For within-quadrant + within-hemisphere and mirror-symmetric, within-quadrant + between-hemisphere comparisons, correlations steadily decreased at larger angular distances suggesting an effect of polar angle distance which likely reflects overlapping response field and homotopic connections, respectively. We found no effect of angular distance between dorsal and ventral V2 and V3 based on actual angular distance or when reflecting across the horizontal (i.e., mirror symmetry) with correlations either remaining about equal to that at iso-polar angle or actually increasing for larger angular distances. For all other bin pairs, there was either little difference in the correlation value at larger angular bin distances or correlations actually got stronger.Author response image 3.Polar Angle-based Intra-run Connectivity.

3) It is also possible that any imbalance in the number of voxels across data bins could bias our results and ability to detect polar angle-based connectivity. To test this, we conducted a new analysis where we equalized the number of nodes in each bin of the topographic model analysis. For bins that contained more than 20 nodes, we subsampled the data to only include 20 nodes. In some subjects, a few bins contained less than 20 nodes. We conducted the new analyses both including and excluding these bins with few nodes. Eccentricity and (lack of) polar angle residual correlations were observed in these new analyses and were consistent with the original results shown in Figure 9. See Figure 13. In some cases, the effects of residual eccentricity correlations were slightly stronger relative to the original analyses (e.g., a few pairs with hV4). Similar results were observed for bins of 30 and 50 nodes as well as whether the bins with few nodes were included or excluded. We now include a description of this control analysis in the Results section.Author response image 4.Equal Bins.

**[Editors' note: further revisions were requested prior to acceptance, as described below.]**

The main issue is that one reviewer is still concerned by the absence of any reflection of polar geometry in the data, and by the apparent new claim that this cannot be due to the quality of your data. During the course of the revision, a second paper with higher quality data has shown that it is possible to see reflections of polar geometry in resting connectivity (Gravel et al. http://journal.frontiersin.org/Journal/10.3389/fnins.2014.00339/).

To the best of our understanding, our results are entirely consistent with those of [41]. We became aware of this study while our manuscript was under re-review. This is an excellent paper demonstrating that retinotopic connectivity (based on overlapping RFs) can be identified in resting state data. We now cite the paper in our Introduction and Discussion sections. In their study, Gravel and colleagues reconstruct polar angle and eccentricity components in areas V2 and V3 for individual subjects based on modeling RF-based connectivity with V1 (referred to as connective field modeling). Consistent with their results, an effect of RF connectivity is qualitatively apparent in the correlation structure of our data (Figure 8), and we found a significant effect of overlapping RF connectivity in our topographic model analysis, which we report in paragraphs 15-17 of Results section. Further, we show in Figure 10 that correlations drop off with angular distance for areas that contain overlapping RFs (e.g., V2v – V3v) as well as for mirror symmetric connections across hemispheres (e.g., V2v RH – V2v LH). In contrast, we do not find clear effects of polar angle connectivity between areas that have minimal / non-overlapping RFs (e.g., V2v – V2d). Gravel and colleagues (2014) did not explore connectivity patterns based on non-overlapping RFs. The polar angle and eccentricity maps that are generated from resting state data in the Gravel study were derived from inter-areal overlapping RF models, and do not account for structure in connectivity between non-overlapping RF regions. To our knowledge, no prior study has systematically explored / shown effects of polar angle (or eccentricity) connectivity attributed to regions with non-overlapping RFs, which is the focus of our manuscript.

We now cite the Gravel paper in the Introduction:

Similarly, fMRI connectivity studies in humans have demonstrated topographically-local correlations between regions with overlapping visual field representations (46; 42; 18; 27; 41; 70).

We also specifically mention the Gravel paper’s results in our Discussion:

By modeling V1 inter-areal connections based on overlapping RFs, Gravel and colleagues (2014) generated polar angle and eccentricity maps for V2 and V3.

*In concert with these changes, we would like you to be slightly clearer about the novelty of your claim with respect to other papers that have shown eccentricity organisation’s – albeit with less detail and precision than you show here*.

*For example, we do not think that the following statement in the Introduction is a fair reflection of the literature*:

*⋖In addition, widespread functional correlation patterns have been observed across visual cortex in both macaques (*[57]; [87]*) and humans (*[65]; [66]; [94]; [27]*). The governing principles, if any, of such widespread connectivity patterns are still unknown.”*

This is a fair point. We agree that the cited statement could be improved to reflect previous literature. We have changed the text to read:

“In addition, widespread functional correlation patterns have been observed across visual cortex in both macaques (57; 87) and humans (65; 66; 94; 27). These patterns contain broad differences between foveal and peripheral cortex (70), though may also be tied to the fine-scale organization of individual retinotopic maps.”

We have also expanded discussion of previous findings and the relation to our findings in the Discussion section.

*Reviewer 3 had related concerns that I reproduce here. We do not require you to perform the analysis in point 1 below, if you are clear (as above) that the polar connectivity can be seen in higher quality data, but if you choose to do the analysis, that would be welcome. However, we would like you to address point 2*.

Reviewer #3:

*I thank the authors for responding my concerns and for adding the analyses shown in Figures 12 and 13. The new results presented in Figure 12, showing that in some cases within and across V2 and V3 (although V1 is not included?) the correlation falls with angular distance, are particularly helpful*.

I am not sure, however, that the concern I had raised about the potential asymmetry in the resolving power along the eccentricity coordinate and polar angle coordinate has been completely addressed. First, the analyses are restricted to only eccentricities from 1 degree to about 12 degrees, so it is not clear how much a given fMRI voxel will sample along the eccentricity direction relative to the polar angle direction given the cortical magnification and therefore how marginal the effect would be. Also, the new analysis accounting for the potential statistical biases arising from re-binning the data does not address the fact that there is higher effective spatial resolution of the eccentricity coordinate than the polar angle coordinate. The plots in Figure 12 are the most direct measure of this, since the observation that in some regions the correlation falls off with distance argues that there may be in some cases enough voxels along the polar angle direction to properly resolve a spatial gradient of correlation. Perhaps the authors could address this issue more directly. (Smoothing the data along the eccentricity direction to yield similar resolution for both directions may be one brute-force approach, but perhaps there is another, better way.)

We appreciate the reviewer’s suggestions regarding the issue of the potential asymmetry in voxel sampling along eccentricity and polar angle dimensions. We believe that our current analyses do address this concern. Our analyses and imaging parameters were sensitive enough for detecting spatial patterns of correlated bold signal along both angular and radial dimensions. We observed significant effects of angular connectivity between regions with overlapping RFs and for mirror symmetrical regions between hemispheres (Figure 6—figure supplement 2, also see Figure 8). These effects are consistent with those previously reported (46; 41; 70). The effects were consistent across binning sizes (i.e., the slopes for individual area pairs did not significantly differ across binning sizes), suggesting that the number of voxels in each bin did not greatly affect our sensitivity for revealing such patterns of connectivity. Thus, a subset of our analyses (Figure 6—figure supplement 2) provides a positive control for our finding of eccentricity-based connectivity in the absence of angular connectivity for regions with non-overlapping RFs (Figure 9).

Any bias in voxel sampling along angular and radial dimensions would affect the (average) signal of individual bins. Since bin pairs, not individual bins, determined the labeling of correlations into overlapping and non-overlapping RF groups (i.e., all bins contributed to both overlapping and non-overlapping RF analyses), any bias would have affected both overlapping and non-overlapping RF analyses. We observed comparable contributions of polar angle and eccentricity predictors for the variance explained by the overlapping RF model in our topographic model analysis (i.e., there was no bias for these comparisons). Due to the topographic nature of cortex, overlapping RF bin pairs will be, on average, in much closer anatomical proximity than non-overlapping RF bin pairs, and thus would actually be more susceptible to any biases driven by lack of sensitivity along the polar angle dimension. Thus, it is unlikely that any such bias in voxel sampling prevented us from resolving effects of polar angle connectivity for non-overlapping RF analyses.

Further, our lab and others (Tootell et al. 1997; Press et al. 2001; Tootell & Hadjikhani 2001; Warnking et al. 2002; Saygin & Sereno 2008; Henriksson et al. 2012) have used comparable voxel resolutions to distinguish polar angle and eccentricity representations across V1-V4 at a finer detail than our binning approach. i.e. each bin contained a range of phase preferences along both polar angle and eccentricity dimensions that can be differentiated in mapping experiments using a voxel resolution (3 – 4 mm) comparable to our current study. As such, our voxel resolution should not have prevented our ability in identifying connectivity effects along eccentricity and polar angle dimensions in the binning approach.

We appreciate the reviewer’s suggestion of asymmetrical smoothing as a control analysis. Given the reasons outlined above, we do not feel this analysis is necessary. Also, such an approach will not directly address the issue for several reasons: (1) Voxel sampling is in 3 dimensions and does not directly translate to biases across the cortical surface along polar angle and eccentricity dimensions, particularly for V2-V4 where cortical folding becomes more complex. (2) Spatial smoothing affects SNR, which could introduce another source of an asymmetrical, eccentricity-based bias. (3) Implementation is a challenge since the asymmetry of the smoothing would have to vary as a function of the cortical magnification and would also vary across subjects. However, our results do hold with the application of a general asymmetrical smoothing similar to the reviewer’s suggestion. To achieve this, we reduced the effective resolution of eccentricity by blurring data across eccentricity bins. We applied a smoothing filter such that a weighted average was computed for the timeseries of each eccentricity bin and all bins at distance of 1 (50% weighting) and 2 (25% weighting) at the iso-polar angle representations. This has the effect of blurring the data along the eccentricity dimension, thereby reducing the effective resolution along that dimension specifically. Even when applying this blurring, we observed eccentricity effects comparable with the original analyses.

As discussed above, the effects of polar angle connectivity reported in previous studies (e.g., [46]; [41]; [70]) are attributed to overlapping RFs. To our knowledge, no study has shown angular-based connectivity patterns between regions with non-overlapping RFs. However, we acknowledge that the lack of observing such connectivity is a negative finding. As with any negative finding, this should not be overstated. Thus, we added the following text to our manuscript in paragraph 11 of Results section:

“No effect of angular distance was observed between dorsal and ventral comparisons based on actual angular distance or when reflecting across the horizontal meridian (i.e., mirror symmetry). As with any negative finding, we cannot conclude with certainty that such angular connectivity does not exist, though, to our knowledge, no study has shown angular-based connectivity patterns between regions with non-overlapping RFs.”

And paragraph 20 of Results section:

“Our data suggests little effect of polar angle connectivity beyond overlapping RFs. As with any negative finding, we can only report that we failed to see an effect of angular connectivity between regions with non-overlapping RFs, but do not conclude with certainty that it does not exist.”

Still, the new results provided in Figure 12 highlight that there is some level of “polar angle connectivity”, which was previously only indirectly demonstrated, e.g., in the second and third panels of Figure 2 and in Figure 8. As the authors point out, based on the topographic model regression analyses there is evidence for “local connectivity patterns” including overlapping RF effects and correlations between homotopic regions. Beneath these overlapping RF and homotopic correlations there are also the widespread correlations, and this pattern appears to be weaker in the polar angle direction. Perhaps the authors would consider including the results of Figure 12 in the main manuscript, and boiling down the results of Figure 9 by reporting the overall variance explained by local connectivity versus widespread connectivity and what proportion of the widespread connectivity is along the eccentricity direction and what proportion is along the polar angle direction. While the authors do point out that these four spatial components are not orthogonal, some clarification of how much of the observed correlations can be explained by widespread patterns would strengthen the manuscript.

We have added Figure 12 into the main manuscript as Figure 6—figure supplement 2 and now discuss polar angle connectivity effects in the Results. We have moved the former Figure 6—figure supplement 2 to Figure 6—figure supplement 3.

We thank the reviewer for the suggestion, and agree that some comparison between the variance explained by local and widespread models is informative. However, a ratio is not ideal for comparing variance explained by the eccentricity and polar angle models since residual polar angle correlations (in contrast to the overlapping RF and eccentricity models) were often very close to zero or negative. We believe that the paired t-test between the residual eccentricity and polar angle correlations paragraphs 17-18 of Results section serves as a better comparison.

We now report the ratio between the variance explained by the local connectivity model and the variance explained by the residual widespread connectivity model in Results section:

“Comparison of local and widespread connectivity influences

The variance explained by the overlapping RF model (after accounting for patterns attributable to the “noise” model) was generally greater than the variance explained by the model of residual widespread connectivity (after removal of overlapping RF connectivity effects) for within hemisphere comparisons (mean ratio across areas, conditions = 1.69:1.00). For between hemisphere comparisons, variance explained by the overlapping RF model (after accounting for patterns attributable to the “noise” model) was generally weaker than the variance explained by the model of residual widespread connectivity without accounting for mirror symmetrical connections (mean ratio across areas, condition = 1.00:5.65), but was generally greater when accounting for these connections (mean ratio across areas, condition = 1.27:1.00).”